# Continual Deep Learning by Functional Regularisation of Memorable Past

**Pingbo Pan,**[1,*,†] **Siddharth Swaroop,**[2,*,‡] **Alexander Immer,**[3,†] **Runa Eschenhagen,**[4,†]
**Richard E. Turner,**[2] **Mohammad Emtiyaz Khan**[5,‡].

[1] University of Technology Sydney, Australia
[2] University of Cambridge, Cambridge, UK
[3] École Polytechnique Fédérale de Lausanne, Switzerland
[4] University of Tübingen, Tübingen, Germany
[5] RIKEN Center for AI Project, Tokyo, Japan

## Abstract

Continually learning new skills is important for intelligent systems, yet standard deep learning methods suffer from catastrophic forgetting of the past. Recent works address this with weight regularisation. Functional regularisation, although computationally expensive, is expected to perform better, but rarely does so in practice. In this paper, we fix this issue by using a new functional-regularisation approach that utilises a few *memorable past* examples crucial to avoid forgetting. By using a Gaussian Process formulation of deep networks, our approach enables training in weight-space while identifying both the memorable past and a functional prior. Our method achieves state-of-the-art performance on standard benchmarks and opens a new direction for life-long learning where regularisation and memory-based methods are naturally combined.

## 1 Introduction

The ability to quickly adapt to changing environments is an important quality of intelligent systems. For such quick adaptation, it is important to be able to identify, memorise, and recall useful past experiences when acquiring new ones. Unfortunately, standard deep-learning methods lack such qualities, and can quickly forget previously acquired skills when learning new ones [18]. Such catastrophic forgetting presents a big challenge for applications such as robotics, where new tasks can appear during training, and data from previous tasks might be unavailable for retraining.

In recent years, many methods have been proposed to address catastrophic forgetting in deep neural networks (DNNs). One popular approach is to keep network weights close to the values obtained for the previous tasks/data [12, 18, 22, 37]. However, this may not always ensure the quality of predictions on previous tasks. Since the network outputs depend on the weights in a complex way, such *weight-regularisation* may not be effective. A better approach is to use *functional-regularisation*, where we directly regularise the network outputs [5], but this is costly because it requires derivatives of outputs at many input locations. Existing approaches reduce these costs by carefully selecting the locations, e.g. by using a *working memory* [5] or Gaussian-Process (GP) inducing points [34], but currently they do not consistently outperform existing weight-regularisation methods.

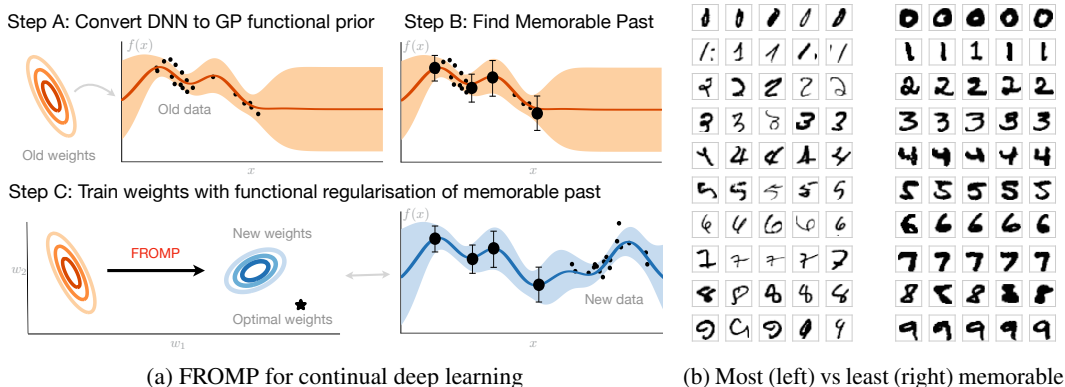

Step A: Convert DNN to GP functional prior    Step B: Find Memorable Past

Step C: Train weights with functional regularisation of memorable past

(a) FROMP for continual deep learning    (b) Most (left) vs least (right) memorable

Figure 1: (a) Our FROMP method consists of three main steps where we convert a DNN to GP using Khan et al. [16], find memorable examples, and train weights with functional regularisation of those examples. (b) Memorable past on MNIST – they are difficult to classify and close to the boundary.

To address this issue, we propose a new functional-regularisation method called Functional Regularisation of Memorable Past (FROMP). Our key idea is to regularise the network outputs at a few *memorable past* examples that are crucial to avoid forgetting. We use a GP formulation of DNNs to obtain a weight-training method that exploits correlations among memorable examples in the function space (see Fig. 1a). FROMP involves a slight modification of Adam and a minor increase in computation cost. It achieves state-of-the-art performance on standard benchmarks, and is consistently better than both the existing weight-regularisation and functional-regularisation methods. Our work in this paper focuses on avoiding forgetting, but it also opens a new direction for life-long learning methods where regularisation methods are naturally combined with memory-based methods.[1]

## 1.1 Related Works

Broadly, existing work on continual learning can be split into three types of approaches: inference-based, memory/rehearsal-based, and model-based. There have also been hybrid approaches attempting to combine them. Inference-based approaches have mostly focused on weight regularisation [2, 9, 12, 18, 22, 37], with some recent efforts on functional regularisation [5, 19, 34]. Our work falls in the latter category, but also imposes functional constraints at datapoints, thereby connecting to memory-based approaches.

Our goal is to consistently outperform weight-regularisation which can be inadequate and brittle for continual deep learning (see Fig. 6 and App. G for an example). The proposed method further addresses many issues with existing functional-regularisation methods [5, 34]. Arguably the work most closely related to ours is the GP-based method of Titsias et al. [34], but there are several key differences. First, our kernel uses *all* the network weights (they use just the last layer) which is important, especially in the early stages of learning when all the weights are changing (see App. G.4 for an example). Second, our functional prior regularises the mean to be close to the past mean, which is lacking in the regulariser of Titsias et al. [34] (see the discussion after Eq. 7). Third, our memorable past examples play a similar role as the inducing inputs, but are much cheaper to obtain (Titsias et al. [34] requires solving a discrete optimisation problem), and have an intuitive interpretation (see Fig. 1b). Due to these differences, our method outperforms the method of Titsias et al. [34], which, unlike ours, performs worse than the weight-regularisation method of Swaroop et al. [33]. We also obtain state-of-the-art performance on a larger Split CIFAR benchmark, a comparison missing in Titsias et al. [34]. Our method is also different to Benjamin et al. [5], which lacks a mechanism to automatically weight past memory and estimate uncertainty.

Our method is based on a set of memorable past examples. Many such memory-based approaches exist. These either maintain a memory of past data examples [9, 22, 25] or train generative models on previous tasks to rehearse pseudo-inputs [30]. Recent work [3, 11] has focused on improving memory-building methods while combining them with inference-based approaches, building on

[1]Code for all experiments is available at `https://github.com/team-approx-bayes/fromp`.

Gradient-Episodic Memory [10, 20]. Compared to these approaches, an advantage of our method is that the memory is obtained within the functional-regularisation framework and does not require solving a separate optimisation problem. The computation is also straightforward, simply requiring a forward-pass through the network followed by picking the top examples (see Sec. 3.2). Finally, model-based approaches change the model architecture during training [13, 27, 29], and this can be combined with other approaches [28]. It is possible to use similar features in our GP-based framework. This is an interesting future direction to be pursued.

## 2 Continual Learning with Weight/Functional Regularisation

In deep learning, we minimise loss functions to estimate network weights. For example, in supervised multi-class classification problems, we are given a dataset $\mathcal{D}$ of $N$ input-output pairs with outputs $\mathbf{y}_i$, a one-hot encoded vector of $K$ classes, and inputs $\mathbf{x}_i$, a vector of length $D$. Our goal is to minimise a loss which takes the following form: $N\bar{\ell}(\mathbf{w}) + \delta R(\mathbf{w})$, where $\bar{\ell}(\mathbf{w}) := \frac{1}{N}\sum_{i=1}^{N}\ell(\mathbf{y}_i, \mathbf{f}_w(\mathbf{x}_i))$ with deep neural network $\mathbf{f}_w(\mathbf{x}) \in \mathbb{R}^K$ and its weights $\mathbf{w} \in \mathbb{R}^P$, $\ell(\mathbf{y}, \mathbf{f})$ denotes a differentiable loss function (e.g., cross entropy) between an output $\mathbf{y}$ and the network output $\mathbf{f}$, $R(\mathbf{w})$ is a regularisation function (usually an $L_2$-regulariser $R(\mathbf{w}) := \mathbf{w}^\top \mathbf{w}$), and $\delta > 0$ controls the regularisation strength. Standard deep-learning approaches rely on an unbiased stochastic gradient of the loss $\bar{\ell}$. This usually requires access to all the data examples for all classes throughout training [8]. It is this unbiased, minibatch setting where deep-learning excels and achieves state-of-the-art performance.

In reality, we do not always have access to all the data at once, and it is not possible to obtain unbiased stochastic gradients. New classes may appear during training and old classes may never be seen again. For such settings, vanilla mini-batch stochastic-gradient methods lead to catastrophic forgetting of past information [18]. Our goal in this paper is to design methods that can avoid, or minimise, such catastrophic forgetting. We focus on a particular setting where the classification task is divided into several tasks, e.g., a task may consist of a classification problem over a subset of classes. We assume that the tasks arrive sequentially one after the other, and task boundaries are provided to us. Once the learning is over, we may never see that task again. Such continual-learning settings have been considered in other works [18, 22, 37] with the goal to avoid forgetting of previous tasks. We also allow storing some past data, which may not always be possible, e.g., due to privacy constraints.

Recent methods have proposed weight-regularisation as a way to combat catastrophic forgetting. The main idea is to find the important weights for past tasks, and keep new weights close to them. For example, when training on the task $t$ while given weights $\mathbf{w}_{t-1}$ trained on the past tasks, we can minimise the following loss: $N\bar{\ell}_t(\mathbf{w}) + \delta(\mathbf{w} - \mathbf{w}_{t-1})^\top \mathbf{F}_{t-1}(\mathbf{w} - \mathbf{w}_{t-1})$, where $\bar{\ell}_t(\mathbf{w})$ is the loss defined over all data examples from task $t$ and $\mathbf{F}_{t-1}$ is a preconditioning matrix that favours the weights relevant to the past tasks more than the rest. The Elastic-Weight Consolidation (EWC) method [18] and Ritter et al. [26], for example, use the Fisher information matrix as the pre-conditioner, while variational continual learning (VCL) [22] employs the precision matrix of the variational approximation. To reduce the computational complexity, it is common to use a diagonal matrix. Such weight-space methods reduce forgetting but do not produce satisfactory results.

The challenge in using weight-regularisation lies in the fact that the exact values of the weights do not really matter due to parametric symmetries [5, 6]. Making current weights closer to the previous ones may not always ensure that the predictions on the past tasks also remain unchanged. Since the network outputs depend on the weights in a complex way, it is difficult to ensure the effectiveness of *weight-regularisation*. A better approach is to directly regularise the outputs, because what ultimately matters is the network output, not the values of the weights. For example, we can use an $L_2$-regulariser over the function values on data examples from past tasks (e.g., see [5]) :

$$\min_w N\bar{\ell}_t(\mathbf{w}) + \delta \sum_{s=1}^{t-1}(\mathbf{f}_{t,s} - \mathbf{f}_{t-1,s})^\top(\mathbf{f}_{t,s} - \mathbf{f}_{t-1,s}), \tag{1}$$

where $\mathbf{f}_{t,s}$ and $\mathbf{f}_{t-1,s}$ are vectors of function values $f_w(\mathbf{x}_i)$ and $f_{w_{t-1}}(\mathbf{x}_i)$ respectively for all $i \in \mathcal{D}_s$ with $\mathcal{D}_s$ being the dataset for the task $s$. Rather than making the weights $\mathbf{w}$ similar to $\mathbf{w}_{t-1}$, such *functional-regularisation* approaches directly force the function values to be similar. Because of this, we expect them to perform better. This is also expected for a Bayesian approach, as posterior approximations in the function-space might be better than those in the weight-space.

Unfortunately, such functional-regularisation is computationally infeasible because it requires us to store all past data and compute function values over them. This computational issue is typically solved by using a subset of inputs. Benjamin et al. [5] employ a *working memory* [20, 25] while Titsias et al. [34] use the inducing point method based on a Gaussian process framework. As discussed earlier, such approaches do not consistently perform better than existing weight-regularisation methods. This could be due to the methods they use to build memory or enforce functional regularisation. Our goal in this paper is to design a functional-regularisation method that is consistently better than weight-regularisation. We build upon the method of Khan et al. [16] to convert deep networks into Gaussian processes, as described next.

## 3 Functional-Regularisation of Memorable Past (FROMP)

### 3.1 From Deep Networks to Functional Priors

Khan et al. [16] propose an approach called DNN2GP to convert deep networks to Gaussian processes (GPs). We employ such GPs as functional priors to regularise the next task. The DNN2GP approach is very similar to the standard weight-space to function-space conversion for linear basis-function models [24]. For example, consider a linear regression model on a scalar output $y_i = f_w(\mathbf{x}_i) + \epsilon_i$ with a function output $f_w(\mathbf{x}_i) := \boldsymbol{\phi}(\mathbf{x}_i)^\top \mathbf{w}$ using a feature map $\boldsymbol{\phi}(\mathbf{x})$. Assume Gaussian noise $\mathcal{N}(\epsilon_i|0, \Lambda^{-1})$ and a Gaussian prior $\mathcal{N}(\mathbf{w}|0, \delta^{-1}\mathbf{I}_P)$ where $\mathbf{I}_P$ is the identity matrix of size $P \times P$. It can then be shown that the posterior distribution of this linear model, denoted by $\mathcal{N}(\mathbf{w}|\mathbf{w}_{\text{lin}}, \boldsymbol{\Sigma}_{\text{lin}})$, induces a GP posterior on function $f_w(\mathbf{x})$ whose mean and covariance functions are given as follows (see App. A.1 or Chapter 2 in Rasmussen and Williams [24]):

$$m_{\text{lin}}(\mathbf{x}) := f_{w_{\text{lin}}}(\mathbf{x}), \quad \kappa_{\text{lin}}(\mathbf{x}, \mathbf{x}') := \boldsymbol{\phi}(\mathbf{x})^\top \boldsymbol{\Sigma}_{\text{lin}} \, \boldsymbol{\phi}(\mathbf{x}'), \tag{2}$$

where $\mathbf{w}_{\text{lin}}$ is simply the Maximum A Posteriori (MAP) estimate of the linear model, and

$$\boldsymbol{\Sigma}_{\text{lin}}^{-1} := \sum_{i=1}^{N} \boldsymbol{\phi}(\mathbf{x}_i) \, \Lambda \, \boldsymbol{\phi}(\mathbf{x}_i)^\top + \delta\mathbf{I}_P. \tag{3}$$

DNN2GP computes a similar GP posterior but for a *neural network* whose posterior is approximated by a Gaussian. Specifically, given a local minimum $\mathbf{w}_*$ of the loss $N\bar{\ell}(\mathbf{w}) + \frac{1}{2}\delta\mathbf{w}^\top \mathbf{w}$ for a scalar output $f_w(\mathbf{x})$, we can construct a Gaussian posterior approximation. Following Khan et al. [16], we employ a variant of the Laplace approximation with mean $\boldsymbol{\mu}_* = \mathbf{w}_*$ and covariance

$$\boldsymbol{\Sigma}_*^{-1} = \sum_{i=1}^{N} \mathbf{J}_{w_*}(\mathbf{x}_i)^\top \Lambda_{w_*}(\mathbf{x}_i, \mathbf{y}_i) \, \mathbf{J}_{w_*}(\mathbf{x}_i) + \delta\mathbf{I}_P, \tag{4}$$

where $\Lambda_{w_*}(\mathbf{x}, \mathbf{y}) := \nabla_{\text{ff}}^2 \ell(\mathbf{y}, \mathbf{f})$ is the scalar Hessian of the loss function, and $\mathbf{J}_{w_*}(\mathbf{x}) := \nabla_w \mathbf{f}_w(\mathbf{x})^\top$ is the $1 \times P$ Jacobian; all quantities evaluated at $\mathbf{w} = \mathbf{w}_*$. Essentially, this variant uses a Gauss-Newton approximation for the covariance instead of the Hessian. Comparing Eqs. 3 and 4, we can interpret $\boldsymbol{\Sigma}_*$ as the covariance of a linear model with a feature map $\boldsymbol{\phi}(\mathbf{x}) = \mathbf{J}_{w_*}(\mathbf{x})^\top$ and noise precision $\Lambda = \Lambda_{w_*}(\mathbf{x}, y)$. Using this similarity, Khan et al. [16] derive a GP posterior approximation for neural networks. They show this for a generic loss function (see App. B2 in their paper), e.g., for a regression loss, the mean and covariance functions of the GP posterior take the following form:

$$m_{w_*}(\mathbf{x}) := f_{w_*}(\mathbf{x}), \quad \kappa_{w_*}(\mathbf{x}, \mathbf{x}') := \mathbf{J}_{w_*}(\mathbf{x}) \, \boldsymbol{\Sigma}_* \, \mathbf{J}_{w_*}(\mathbf{x}')^\top. \tag{5}$$

A similar equation holds for other loss functions such as those used for binary and multiclass classification; see App. A.2 for details. We denote such GP posteriors by $\mathcal{GP}(m_{w_*}(\mathbf{x}), \kappa_{w_*}(\mathbf{x}, \mathbf{x}'))$, and use them as a *functional prior* to regularise the next task.

The above result holds at a minimiser $\mathbf{w}_*$, but can be extended to a sequence of weights obtained during optimisation [16]. For example, for Gaussian variational approximations $q(\mathbf{w})$, we can obtain GP posteriors by replacing $\mathbf{w}_*$ by a sample $\mathbf{w} \sim q(\mathbf{w})$ in Eq. 5. We denote such GPs by $\mathcal{GP}(m_w(\mathbf{x}), \kappa_w(\mathbf{x}, \mathbf{x}'))$. The result also applies to variants of Newton's method, RMSprop, and Adam (see App. A.3). As shown in [16], many DNN2GP posteriors are related to the Neural Tangent Kernel (NTK) [14], e.g., the prior distribution to obtain the posterior in Eq. 5 corresponds to the NTK of a finite-width network. A slightly different kernel is obtained when a variational approximation is used. Unlike the method of Titsias et al. [34], the kernel above uses *all* the network weights, and uses the Jacobians instead of the network output or its last layer.

## 3.2 Identifying Memorable past

To reduce the computation cost of functional regularisation, we identify a few memorable past examples. To do so, we exploit a property of linear models. Consider a linear model where different noise precision $\Lambda_i$ is assigned to each pair $\{\mathbf{x}_i, y_i\}$. For MAP estimation, the examples with high value of $\Lambda_i$ contribute more, as is clear from the objective: $\mathbf{w}_{\text{MAP}} = \arg\max_w \sum_{i=1}^{N} \Lambda_i (y_i - \phi(\mathbf{x}_i)^\top \mathbf{w})^2 + \delta \mathbf{w}^\top \mathbf{w}$. The noise precision $\Lambda_i$ can therefore be interpreted as the relevance of the data example $i$. Such relevant examples are crucial to ensure that the solution stays at $\mathbf{w}_{\text{MAP}}$ or close to it. These ideas are widely used in the theory of leverage-score sampling [1, 21] to identify the most *influential* examples. Computation using such methods is infeasible since they require inverting a large matrix. Titsias et al. [34] use an approximation by inverting smaller matrices, but they require solving a discrete optimisation problem to select examples. We propose a method which is not only cheap and effective, but also yields intuitive results.

We use the linear model corresponding to the GP posterior from Section 3.1. The linear model assigns different noise precision to each data example. See Eqs. 3 and 4 where the quantity $\Lambda_{w_*}(\mathbf{x}_i, y_i)$ plays the same role as the noise precision $\Lambda$. Therefore, $\Lambda_{w_*}(\mathbf{x}_i, y_i)$ can be used as a relevance measure, and a simple approach to pick influential examples is to sort it $\forall i$ and pick the top few examples. We refer to such a set of examples as the *memorable past* examples. An example is shown in Fig. 1b where our approach picks many examples that are difficult to classify. The memorable past can be intuitively thought of as *examples close to the decision boundary*. An advantage of using this approach is that $\Lambda_{w_*}(\mathbf{x}_i, y_i)$ is extremely cheap to compute. It is simply the second derivative of the loss, which can be obtained with a forward pass to get $\ell(y_i, \hat{y}_i)$, followed by double differentiation with respect to $\hat{y}_i$. For binary classification, our approach is equivalent to the "Confidence Sampling" approaches used in the Active Learning literature [4, 36], although in general it differs from them. After training on task $t$, we select a set of few memorable examples in $\mathcal{D}_t$, which we denote by $\mathcal{M}_t$.

## 3.3 Training in weight-space with a functional prior

We will now describe the final step for weight-training with functional-regularisation. We use the Bayesian formulation of continual learning and replace the prior distribution in weight space by a *functional prior*. Given a loss of the form $N\bar{\ell}_t(\mathbf{w}) + R(\mathbf{w})$, a Bayesian formulation in weight-space employs a regulariser that uses the previous posterior, i.e., $R(\mathbf{w}) \equiv -\log p(\mathbf{w}|\mathcal{D}_{1:t-1})$. Computing the exact posterior, or a tempered version of it, would in theory avoid catastrophic forgetting, but that is expensive and we must use approximations. For example, Nguyen et al. [22] use the variational approximation from the previous task $p(\mathbf{w}|\mathcal{D}_{1:t-1}) \approx q_{t-1}(\mathbf{w}) = \mathcal{N}(\mathbf{w}|\boldsymbol{\mu}, \boldsymbol{\Sigma})$ to obtain the weight regulariser. Our goal is to replace such weight regulariser by a functional regulariser obtained by using the GP posteriors described in Sec. 3.1.

We use functional regularisation defined over memorable examples. Denote by $\mathbf{f}$ the vector of function values defined at all memorable past $\mathcal{M}_s$ in all tasks $s < t$. Denoting a sample from $q(\mathbf{w})$ by $\mathbf{w}$, we can obtain a GP posterior over $\mathbf{f}$ by using Eq. 5. We denote it by $\tilde{q}_w(\mathbf{f}) = \mathcal{N}(\mathbf{f}|\mathbf{m}_t(\mathbf{w}), \mathbf{K}_t(\mathbf{w}))$, where $\mathbf{m}_t(\mathbf{w})$ and $\mathbf{K}_t(\mathbf{w})$ respectively denote the mean vector and kernel matrix obtained by evaluating $\mathcal{GP}(m_w(\mathbf{x}), \kappa_w(\mathbf{x}, \mathbf{x}'))$ at the memorable past examples. Similarly, denoting a sample from $q_{t-1}(\mathbf{w})$ by $\mathbf{w}_{t-1}$, we can obtain another GP posterior, which we call the *functional prior*, denoted by $\tilde{q}_{w_{t-1}}(\mathbf{f}) = \mathcal{N}(\mathbf{f}|\mathbf{m}_{t-1}, \mathbf{K}_{t-1})$. Using these two GPs, we can replace the weight regulariser used in [22] by a *functional regulariser* which is equal to the expectation of the functional prior:

$$\min_{q(w)} \mathbb{E}_{q(w)} \left[ (N/\tau)\bar{\ell}_t(\mathbf{w}) + \log q(\mathbf{w}) \right] - \underbrace{\mathbb{E}_{q(w)} \left[ \log q_{t-1}(\mathbf{w}) \right]}_{\approx \mathbb{E}_{\tilde{q}_w(\mathbf{f})} \left[ \log \tilde{q}_{w_{t-1}}(\mathbf{f}) \right]}, \tag{6}$$

where the last term is the weight regulariser, and $\tau > 0$ is a tempering parameter. Fortunately, the functional regulariser has a closed-form expression: $\mathbb{E}_{\tilde{q}_w(\mathbf{f})} \left[ \log \tilde{q}_{w_{t-1}}(\mathbf{f}) \right] =$

$$-\tfrac{1}{2} \left[ \text{Tr}(\mathbf{K}_{t-1}^{-1}\mathbf{K}_t(\mathbf{w})) + (\mathbf{m}_t(\mathbf{w}) - \mathbf{m}_{t-1})^\top \mathbf{K}_{t-1}^{-1} (\mathbf{m}_t(\mathbf{w}) - \mathbf{m}_{t-1}) \right] + \text{constant}. \tag{7}$$

This term depends on $\boldsymbol{\mu}$ and $\boldsymbol{\Sigma}$ through the sample $\mathbf{w} \sim q(\mathbf{w})$. The regulariser is an approximation for reasons discussed in App. D. The regulariser has a similar form[2] to Titsias et al. [34], but unlike their

**Algorithm 1:** FROMP for binary classification on task $t$ given $q_{t-1}(\mathbf{w}) := \mathcal{N}(\boldsymbol{\mu}_{t-1}, \mathrm{diag}(\mathbf{v}_{t-1}))$, and memorable pasts $\mathcal{M}_{1:t-1}$. Additional computations on top of Adam are highlighted in red.

---

**Function** `FROMP`$(\mathcal{D}_t, \boldsymbol{\mu}_{t-1}, \mathbf{v}_{t-1}, \mathcal{M}_{1:t-1})$:
  Get $\mathbf{m}_{t-1,s}, \mathbf{K}_{t-1,s}^{-1}, \forall$ tasks $s < t$ (Eq. 10)
  Initialise $\mathbf{w} \leftarrow \boldsymbol{\mu}_{t-1}$
  **while** *not converged* **do**
      Randomly sample $\{\mathbf{x}_i, y_i\} \in \mathcal{D}_t$
      $\mathbf{g} \leftarrow N \nabla_w \ell(y_i, f_w(\mathbf{x}_i))$
      $\mathbf{g}_f \leftarrow$ `g_FR`$(\mathbf{w}, \mathbf{m}_{t-1}, \mathbf{K}_{t-1}^{-1}, \mathcal{M}_{1:t-1})$
      Adam update with gradient $\mathbf{g} + \tau \mathbf{g}_f$
  **end**
  $\boldsymbol{\mu}_t \leftarrow \mathbf{w}$ and compute $\mathbf{v}_t$ (Eq. 9)
  $\mathcal{M}_t \leftarrow$ `memorable_past`$(\mathcal{D}_t, \mathbf{w})$
  **return** $\boldsymbol{\mu}_t, \mathbf{v}_t, \mathcal{M}_t$

**Function** `g_FR`$(\mathbf{w}, \mathbf{m}_{t-1}, \mathbf{K}_{t-1}^{-1}, \mathcal{M}_{1:t-1})$:
  Initialise $\mathbf{g}_f \leftarrow \mathbf{0}$
  **for** *task* $s = 1, 2, ..., t-1$ **do**
      Compute $\mathbf{m}_{t,s}$ (Eq. 10)
      $\mathbf{h}_i \leftarrow \Lambda_w(\mathbf{x}_i) \mathbf{J}_w(\mathbf{x}_i)^\top, \forall \mathbf{x}_i \in \mathcal{M}_s$
      Form matrix $\mathbf{H}$ with $\mathbf{h}_i$ as columns
      $\mathbf{g}_f \leftarrow \mathbf{g}_f + \mathbf{H}\mathbf{K}_{t-1,s}^{-1}(\mathbf{m}_{t,s} - \mathbf{m}_{t-1,s})$
  **end**
  **return** $\mathbf{g}_f$
**Function** `memorable_past`$(\mathcal{D}_t, \mathbf{w})$:
  Calculate $\Lambda_w(\mathbf{x}_i), \ \forall \mathbf{x}_i \in \mathcal{D}_t$.
  **return** $M$ examples with highest $\Lambda_w(\mathbf{x}_i)$.

---

regulariser, ours forces the mean $\mathbf{m}_t(\mathbf{w})$ to be close to $\mathbf{m}_{t-1}$, which is desirable since it encourages the predictions of the past to remain the same.

Optimising $\boldsymbol{\mu}$ and $\boldsymbol{\Sigma}$ in Eq. 6 with this functional prior can be very expensive for large networks. We make five approximations to reduce the cost, discussed in detail in App. B. First, for the functional prior, we use the mean of $q_{t-1}(\mathbf{w})$, instead of a sample $\mathbf{w}_{t-1}$, which corresponds to using the GP posterior of Eq. 5. Second, for Eq. 7, we ignore the derivative with respect to $\mathbf{K}_t(\mathbf{w})$ and only use $\mathbf{m}_t(\mathbf{w})$, which assumes that the Jacobians do not change significantly. Third, instead of using the full $\mathbf{K}_{t-1}$, we factorise it across tasks, i.e., let it be a block-diagonal matrix with $\mathbf{K}_{t-1,s}, \forall s$ as the diagonal. This makes the cost of inversion linear in the number of tasks. Fourth, following Khan et al. [16], we propose to use a deterministic optimiser for Eq. 6, which corresponds to setting $\mathbf{w} = \boldsymbol{\mu}$. Finally, we use a diagonal $\boldsymbol{\Sigma}$, which corresponds to a mean-field approximation, reducing the cost of inversion. As shown in App. B, the resulting algorithm finds a solution to the following problem:

$$\min_w N\bar{\ell}_t(\mathbf{w}) + \tfrac{1}{2}\tau \sum_{s=1}^{t-1} \left[\mathbf{m}_{t,s}(\mathbf{w}) - \mathbf{m}_{t-1,s}\right]^\top \mathbf{K}_{t-1,s}^{-1} \left[\mathbf{m}_{t,s}(\mathbf{w}) - \mathbf{m}_{t-1,s}\right], \qquad (8)$$

where $\mathbf{m}_{t,s}$ is the sub-vector of $\mathbf{m}_t$ corresponding to the task $s$. The above is a computationally-cheap approximation of Eq. 6 and forces the network to produce similar outputs at memorable past examples. The objective is an improved version of Eq. 1 [5]. For regression, the mean $\mathbf{m}_{t,s}$ in Eq. 8 is equal to the vector $\mathbf{f}_{t,s}$ used in Eq. 1. Our functional regulariser additionally includes a kernel matrix $\mathbf{K}_{t-1,s}$ to take care of the uncertainty and weighting of past tasks' memorable examples.

Due to a full kernel matrix, the functional regulariser exploits the correlations between memorable examples. This is in contrast with a weight-space approach, where modelling correlations is infeasible since $\boldsymbol{\Sigma}$ is extremely large. Here, training is cheap since the objective in Eq. 8 can be optimised by using Adam. *Our approach therefore provides a cheap weight-space training method while exploiting correlations in function-space.* Due to these properties, we expect our method to perform better. We can expect further improvements by relaxing these assumptions (see App. B), e.g., we can use a full kernel matrix, use a variational approximation, or employ a block-diagonal covariance matrix. We leave such comparisons as future work since they require sophisticated implementation to scale.

### 3.4 The final algorithm and computational complexity

The resulting algorithm, FROMP, is shown in Alg. 1 for binary classification (extension to multiclass classification is in App. C). For binary classification, we assume a sigmoid $\sigma(f_w(\mathbf{x}))$ function and cross-entropy loss. As shown in App. A.2, the Jacobian (of size $1 \times P$) and noise precision (a scalar) are as follows: $\mathbf{J}_w(\mathbf{x}) = \nabla_w f_w(\mathbf{x})^\top$ and $\Lambda_w(\mathbf{x}) = \sigma(f_w(\mathbf{x}))[1 - \sigma(f_w(\mathbf{x}))]$. To compute the mean and kernel, we need the diagonal of the covariance, which we denote by $\mathbf{v}$. This can be obtained using Eq. 4 but with the sum over $\mathcal{D}_{1:t}$. The update below computes this recursively:

$$\frac{1}{\mathbf{v}_t} = \left[ \frac{1}{\mathbf{v}_{t-1}} + \sum_{i \in \mathcal{D}_t} \mathrm{diag}\left(\mathbf{J}_w(\mathbf{x}_i)^\top \Lambda_w(\mathbf{x}_i)\mathbf{J}_w(\mathbf{x}_i)\right) \right], \qquad (9)$$

where '/' denotes element-wise division and $\mathrm{diag}(\mathbf{A})$ is the diagonal of $\mathbf{A}$. Using this in an expression similar to Eq. 5, we can compute the mean and kernel matrix (see App. A.2 for details):

$$\mathbf{m}_{t,s}(\mathbf{w})[i] = \sigma\left(f_w(\mathbf{x}_i)\right), \quad \mathbf{K}_{t,s}(\mathbf{w})[i,j] = \Lambda_w(\mathbf{x}_i)\left[\mathbf{J}_w(\mathbf{x}_i)\,\mathrm{Diag}\left(\mathbf{v}_t\right)\mathbf{J}_w(\mathbf{x}_j)^\top\right]\Lambda_w(\mathbf{x}_j), \quad (10)$$

over all memorable examples $\mathbf{x}_i, \mathbf{x}_j$, where $\mathrm{Diag}(\mathbf{a})$ denotes a diagonal matrix with $\mathbf{a}$ as the diagonal. Using these, we can write the gradient of Eq. 8, where the gradient of the functional regulariser is added as an additional term to the gradient of the loss: $N\nabla_w\bar{\ell}_t(\mathbf{w}) + \tau\sum_{s=1}^{t-1}(\nabla_w\mathbf{m}_{t,s}(\mathbf{w}))\mathbf{K}_{t-1,s}^{-1}(\mathbf{m}_{t,s}(\mathbf{w}) - \mathbf{m}_{t-1,s})$ where $\nabla_w\mathbf{m}_{t,s}(\mathbf{w})[i] = \nabla_w\left[\sigma\left(f_w(\mathbf{x}_i)\right)\right] = \Lambda_w(\mathbf{x}_i)\mathbf{J}_w(\mathbf{x}_i)^\top$. The regulariser is computed in subroutine g_FR in Alg. 1.

The additional computations on top of Adam are highlighted in red in Alg. 1. Every iteration requires functional gradients (in g_FR) whose cost is dominated by the computation of $\mathbf{J}_w(\mathbf{x}_i)$ at all $\mathbf{x}_i \in \mathcal{M}_s, \forall s < t$. Assuming the size of the memorable past is $M$ per task, this adds an additional $\mathcal{O}(MPt)$ computation, where $P$ is the number of parameters and $t$ is the task number. This increases only linearly with the size of the memorable past. We need three additional computations but they are required only *once per task*. First, inversion of $\mathbf{K}_s, \forall s < t$, which has cost $O(M^3t)$. This is linear in number of tasks and is feasible when $M$ is not too large. Second, computation of $\mathbf{v}_t$ in Eq. 9 requires a full pass through the dataset $\mathcal{D}_t$, with cost $O(NP)$ where $N$ is the dataset size. This cost can be reduced by estimating $\mathbf{v}_t$ using a minibatch of data (as is common for EWC [18]). Finally, we find the memorable past $\mathcal{M}_t$, requiring a forward pass followed by picking the top $M$ examples.

## 4 Experiments

To identify the benefits of the functional prior (step A) and memorable past (step B), we compare FROMP to three variants: (1) FROMP-$L_2$ where we replace the kernel in Eq. 5 by the identity matrix, similar to Eq. 1, (2) FRO*R*P where memorable examples selected randomly ("R" stands for random), (3) FRO*R*P-$L_2$ which is same as FRO*R*P, but the kernel in Eq. 5 is replaced by the identity matrix. We present comparisons on four benchmarks: a toy dataset, permuted MNIST, Split MNIST, and Split CIFAR (a split version of CIFAR-10 & CIFAR-100). Results for the toy dataset are summarised in Fig. 5 and App. G, where we also visually show the brittleness of weight-space methods. In all experiments, we use the Adam optimiser [17]. Details on hyperparameter settings are in App. F.

### 4.1 Permuted and Split MNIST

Permuted MNIST consists of a series of tasks, with each applying a fixed permutation of pixels to the entire MNIST dataset. Similarly to previous work [18, 37, 22, 34], we use a fully connected single-head network with two hidden layers, each consisting of 100 hidden units. We train for 10 tasks. The number of memorable examples is set in the range 10–200. We also test on the Split MNIST benchmark [37], which consists of five binary classification tasks built from MNIST: 0/1, 2/3, 4/5, 6/7, and 8/9. Following the settings of previous work, we use a fully connected multi-head network with two hidden layers, each with 256 hidden units. We select 40 memorable points per task.

The final average accuracy is shown in Fig. 2a where FROMP achieves better performance than weight-regularisation methods (EWC, VCL, SI) as well as the function-regularisation continual learning (FRCL) method [34]. FROMP also improves over FRO*R*P-$L_2$ and FROMP-$L_2$, demonstrating the effectiveness of the kernel. The improvement compared to FRO*R*P is not significant. We believe this is because a random memorable past already achieves a performance close to the highest achievable performance, and we see no further improvement by choosing the examples carefully. However, as shown in Fig. 3c, we do see an improvement when the number of memorable examples are small (compare FRO*R*P vs FROMP). Finally, Fig. 1b shows the most and least memorable examples chosen by sorting $\Lambda_w(\mathbf{x}, y)$. The most memorable examples appear to be more difficult to classify than the least memorable examples, which suggests that they may lie closer to the decision boundary.

We also run FROMP on Split MNIST on a smaller network architecture [33], obtaining $(99.2\pm0.1)\%$ (see App. F.2). Additionally, in App. H, we show that, when the task-boundary information is unavailable, it is still possible to automatically detect the boundaries within our method. When new tasks are encountered, we expect the prediction using current network and past ones to be similar. We use this to detect task boundaries by performing a statistical test; see App. H for details.

| Method | Permuted | Split |
|---|---|---|
| DLP [32] | 82% | 61.2% |
| EWC [18] | 84% | 63.1% |
| SI [37] | 86% | 98.9% |
| Improved VCL [33] | $93 \pm 1\%$ | $98.4 \pm 0.4\%$ |
| + random Coreset | $\mathbf{94.6} \pm 0.3\%$ | $98.2 \pm 0.4\%$ |
| FRCL-RND [34] | $94.2 \pm 0.1\%$ | $97.1 \pm 0.7\%$ |
| FRCL-TR [34] | $94.3 \pm 0.2\%$ | $97.8 \pm 0.7\%$ |
| FRORP-$L_2$ | $87.9 \pm 0.7\%$ | $98.5 \pm 0.2\%$ |
| FROMP-$L_2$ | $94.6 \pm 0.1\%$ | $98.7 \pm 0.1\%$ |
| FRORP | $94.6 \pm 0.1\%$ | $\mathbf{99.0} \pm 0.1\%$ |
| FROMP | $\mathbf{94.9} \pm 0.1\%$ | $\mathbf{99.0} \pm 0.1\%$ |

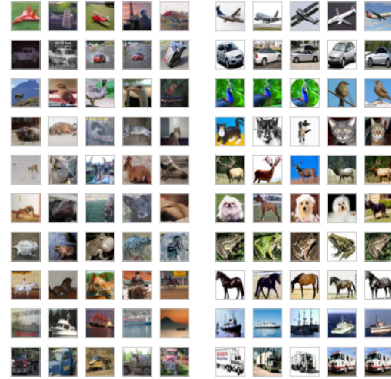

(a) MNIST comparisons: for Permuted, we use 200 examples as memorable/coreset/inducing points. For Split, we use 40.

(b) Most (left) vs least (right) memorable

Figure 2: (a) On MNIST, FROMP obtains better accuracy than weight-regularisation (EWC, SI, VCL) and functional-regularisation (FRCL). Note that FRCL does not outperform 'Improved VCL + random coreset' while FROMP does. The standard errors are reported over 5 runs.

## 4.2 Split CIFAR

Split CIFAR is a more difficult benchmark than MNIST, and consists of 6 tasks. The first task is the full CIFAR-10 dataset, followed by 5 tasks, each consisting of 10 consecutive classes from CIFAR-100. We use the same model architecture as Zenke et al. [37]: a multi-head CNN with 4 convolutional layers, then 2 dense layers with dropout. The number of memorable examples is set in the range 10–200, and we run each method 5 times. We compare to two additional baselines. The first baseline consists of networks trained on each task *separately*. Such training cannot profit from forward/backward transfer from other tasks, and sets a lower limit which we must outperform. The second baseline is where we train all tasks *jointly*, which would yield perhaps the best results and which we would like to match.

The results are summarised in Fig. 3a, where we see that FROMP is close to the upper limit while outperforming all the other methods. The weight-regularisation methods EWC and SI do not perform well on the later tasks while VCL forgets the earlier tasks. Poor performance of VCL is most likely due to the difficulty of using Bayes By Backprop [7] on CNNs[3] [23, 31]. FROMP performs consistently better across all tasks (except the first task where it is close to the best). It also improves over the lower limit ('separate tasks') by a large margin. In fact, on tasks 4-6, FROMP matches the performance to the network trained jointly on all tasks, which implies that there it completely avoids forgetting. The average performance over all tasks is also the best (see the 'Avg' column).

Fig. 3b shows the performance with respect to the number of memorable past examples. Similarly to Fig. 3c, carefully selecting memorable example improves the performance, especially when the number of memorable examples is small. For example, with 10 such memorable examples, a careful selection in FROMP increases the average accuracy to 70% from 45% obtained by FRORP. Including the kernel in FROMP here unfortunately does not improve significantly over FROMP-$L_2$, unlike the MNIST experiment. Fig. 2b shows a few images with most and least memorable past examples where we again see that the most memorable might be more difficult to classify.

Finally, we analyse the forward and backward transfer obtained by FROMP. Forward transfer means the accuracy on the *current* tasks increases as number of past tasks increases, while backward transfer means the accuracy on the *previous* tasks increases as more tasks are observed. As discussed in App. E, we find that, for Split CIFAR, FROMP's forward transfer is much better than VCL and EWC, while its backward transfer is comparable to EWC. We define a forward transfer metric as the average improvement in accuracy on a new task over a model trained *only* on that task (see App. E for an expression). A higher value is better and quantifies the performance gain by observing past tasks. FROMP achieves $6.1 \pm 0.7\%$, a much higher value compared to $0.17 \pm 0.9\%$ obtained with EWC and $1.8 \pm 3.1\%$ with VCL+coresets. For backward transfer, we used the BWT metric defined in Lopez-Paz and Ranzato [20] which roughly captures the difference in accuracy obtained when a task

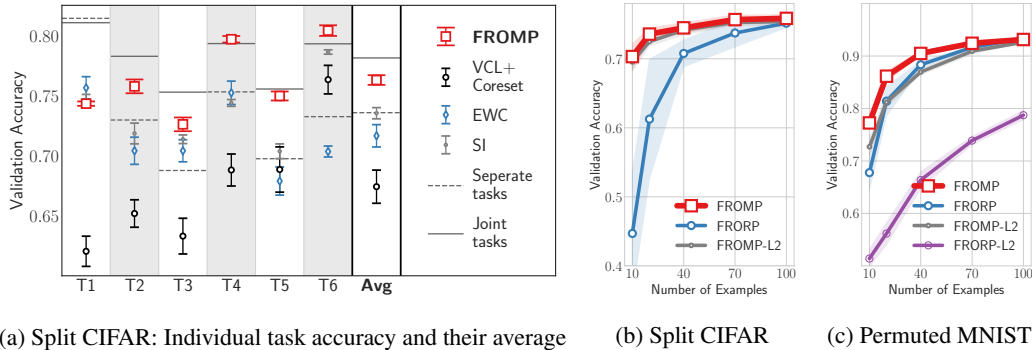

(a) Split CIFAR: Individual task accuracy and their average     (b) Split CIFAR     (c) Permuted MNIST

Figure 3: Fig. (a) shows that FROMP outperforms weight-regularisation methods (see App. F.3 for numerical values). 'Tx' means Task x. Figs. (b) and (c) show average accuracy with respect to the number of memorable examples. A careful selection of memorable examples in FROMP gives better results than random examples in FRORP, especially when the memory size is small. For MNIST, the kernel in FROMP improves performance over FROMP-L2, which does not use a kernel.

is first trained and its accuracy after the final task. Again, higher is better and quantifies the gain obtained with the future tasks. Here, FROMP has a score of $-2.6 \pm 0.9\%$, which is comparable to EWC's score of $-2.3 \pm 1.4\%$ but better than VCL+coresets which obtains $-9.2 \pm 1.8\%$. Performance of FROMP is summarised in Table 1.

| Split MNIST | | Permuted MNIST | | Split CIFAR | |
|---|---|---|---|---|---|
| FWD | BWD | FWD | BWD | FWD | BWD |
| $-0.07 \pm 0.05\%$ | $-0.5 \pm 0.2\%$ | $-1.9 \pm 0.1\%$ | $-1.0 \pm 0.1\%$ | $6.1 \pm 0.7\%$ | $-2.6 \pm 0.9\%$ |

Table 1: Forward and Backward transfer metrics (see App. E for precise definitions and more results) for FROMP on benchmarks. Higher is better.

## 5   Discussion

We propose FROMP, a functional-regularisation approach for continual learning while avoiding catastrophic forgetting. FROMP uses a Gaussian Process formulation of neural networks to convert weight-space distributions into function-space. With this formulation, we proposed ways to identify relevant *memorable past* examples, and functionally regularise the training of neural network weights. FROMP achieves state-of-the-art performance across benchmarks.

This paper takes the first step in a direction to combine the ideas from neural network and GP communities, while maintaining the simplicity of the training methods. There are plenty of future directions to explore. Would using VI instead of a Laplace approximation result in better accuracy? What are some ways to choose memorable examples? Is there a common principle behind them? How many memorable examples should one use, and how can we ensure that increasing their numbers substantially increases the performance? Do we obtain improvements when we relax some assumptions, and what kind of improvements? Will this approach work at large scale, e.g., on the ImageNet dataset? Are there better methods to automatically detect task boundaries? And finally how can all of these ideas fit in a Bayesian framework, and can we obtain theoretical guarantees for such methodologies?

These are some of the questions of future interest. We believe that functional regularisation is ultimately how we want to train deep learning algorithms. We hope that the methods discussed in this paper open new methodologies for knowledge transfer in deep learning.

## Broader Impact

The focus of this paper is on continual deep learning which is related to the field of *life-long learning*. Designing such algorithms is a bottleneck for deep learning which heavily relies on the *offline* setting where all the data is available at once. Life-long learning methods, such as ours, will extend the application of deep learning to problems where data is limited and needs to be collected slowly over time. This could bring a positive change in fields such as robotics, medicine, healthcare, and climate science. A shortcoming currently is the lack of theoretical guarantees, which is essential to ensure a positive change. Life-long learning methods, such as ours, should not be applied to mission-critical problems, until such guarantees are available.

One could imagine negative outcomes too, e.g., if life-long learning methods are perfected, machines could then learn in a sequential fashion, similar to living beings and humans. It is possible that their learning will catch up with ours, which will have a huge affect on the society and economy. We do not see this happening any time soon, and in the short term we see a net positive effect on the society. It is important to perform research to understand effects on society in case life-long learning methods are successful.

## Acknowledgements

Pingbo Pan would like to thank Prof Yi Yang (The ReLER Lab, University of Technology, Sydney) for helpful discussions. Runa Eschenhagen is thankful to Rio Yokota for hosting him in Tokyo Institute of Technology during this work. We are also thankful for the RAIDEN computing system and its support team at the RIKEN Center for AI Project, which we used extensively for our experiments. Siddharth Swaroop is supported by an EPSRC DTP studentship. Richard E. Turner is supported by Google, Amazon, ARM, Improbable, EPSRC grants EP/M0269571 and EP/L000776/1, and the UKRI Centre for Doctoral Training in the Application of Artificial Intelligence to the study of Environmental Risks (AI4ER). Mohammad Emtiyaz Khan is partially supported by KAKENHI Grant-in-Aid for Scientific Research (B), Research Project Number 20H04247.

## Footnotes

\* These two authors contributed equally.

† This work is conducted during an internship at RIKEN Center for AI project, Tokyo, Japan.

‡ Corresponding authors: `emtiyaz.khan@riken.jp`, `ss2163@cam.ac.uk`

[2]Their regulariser is $\mathbb{E}_{q(u_{t-1})}[\log p_w(\mathbf{u}_{t-1})] = -\tfrac{1}{2}\{\text{Tr}[\mathbf{K}(\mathbf{w})^{-1}\boldsymbol{\Sigma}_{t-1}] + \boldsymbol{\mu}_{t-1}^\top \mathbf{K}(\mathbf{w})^{-1}\boldsymbol{\mu}_{t-1}\}$ where $p_w(\mathbf{u}_{t-1}) = \mathcal{N}(0, K(\mathbf{w}))$ with the kernel evaluated at inducing inputs $\mathbf{u}_{t-1}$ and $q(\mathbf{u}_{t-1}) = \mathcal{N}(\boldsymbol{\mu}_{t-1}, \boldsymbol{\Sigma}_{t-1})$. This regulariser encourages $\mathbf{K}(\mathbf{w})$ to remain close to the second moment $\boldsymbol{\Sigma}_{t-1} + \boldsymbol{\mu}_{t-1}\boldsymbol{\mu}_{t-1}^\top$.

[3]Previous results by Nguyen et al. [22] and Swaroop et al. [33] are obtained using multi-layer perceptrons.

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
