[Supplementary Material]

# A  Deep Networks to Functional Priors with DNN2GP

## A.1  GP posteriors from the Minimiser of Linear Model

The posterior distribution of a linear model induces a GP posterior as shown by Rasmussen and Williams [24]. We discuss this in detail now for the following linear model discussed in Sec. 3.1:

$$y_i = f_w(\mathbf{x}_i) + \epsilon_i, \quad \text{where } f_w(\mathbf{x}_i) := \phi(\mathbf{x}_i)^\top \mathbf{w}, \ \epsilon_i \sim \mathcal{N}(\epsilon_i | 0, \Lambda^{-1}), \ \text{and } \mathbf{w} \sim \mathcal{N}(\mathbf{w} | 0, \delta^{-1} \mathbf{I}_P) \tag{11}$$

with a feature map $\phi(\mathbf{x})$. Rasmussen and Williams [24] show that the predictive distribution for a test input $\mathbf{x}$ takes the following form (see Equation 2.11 in their book):

$$p(f(\mathbf{x})|\mathbf{x}, \mathcal{D}) = \mathcal{N}(f(\mathbf{x}) \,|\, \Lambda \phi(\mathbf{x})^\top \mathbf{A}^{-1} \mathbf{\Phi} \mathbf{y}, \ \phi(\mathbf{x})^\top \mathbf{A}^{-1} \phi(\mathbf{x})),$$
$$\text{where } \mathbf{A} := \sum_i \phi(\mathbf{x}_i) \, \Lambda \, \phi(\mathbf{x}_i)^\top + \delta \mathbf{I}_P. \tag{12}$$

where $\mathcal{D}$ is set of training points $\{y_i, \mathbf{x}_i\}$ for $i$, and $\mathbf{\Phi}$ is a matrix with $\phi(\mathbf{x}_i)$ as columns.

Rasmussen and Williams [24] derive the above predictive distribution by using the weight-space posterior $\mathcal{N}(\mathbf{w}|\mathbf{w}_{\text{lin}}, \mathbf{\Sigma}_{\text{lin}})$ with the mean and covariance defined as below:

$$\mathbf{w}_{\text{lin}} := \Lambda \mathbf{A}^{-1} \mathbf{\Phi} \mathbf{y}, \quad \mathbf{\Sigma}_{\text{lin}} := \mathbf{A}^{-1}. \tag{13}$$

The mean $\mathbf{w}_{lin}$ is also the minimiser of the least-squares loss and $\mathbf{A}$ is the hessian at that solution.

Rasmussen and Williams [24] show that the predictive distribution in Eq. 12 corresponds to a GP posterior with the following mean and covariance functions:

$$m_{\text{lin}}(\mathbf{x}) = \Lambda \phi(\mathbf{x})^\top \mathbf{A}^{-1} \mathbf{\Phi} \mathbf{y} = \phi(\mathbf{x})^\top \mathbf{w}_{\text{lin}} = f_{w_{\text{lin}}}(\mathbf{x}), \tag{14}$$
$$\kappa_{\text{lin}}(\mathbf{x}, \mathbf{x}') := \phi(\mathbf{x})^\top \mathbf{\Sigma}_{\text{lin}} \, \phi(\mathbf{x}'), \tag{15}$$

This is the result shown in Eq. 2 in Sec. 3.1. We can also write the predictive distribution of the observation $y = f(\mathbf{x}) + \epsilon$ where $\epsilon \sim \mathcal{N}(0, \Lambda^{-1})$ as follows:

$$p(y|\mathbf{x}, \mathcal{D}) = \mathcal{N}(y \,|\, \underbrace{f_{w_{\text{lin}}}(\mathbf{x})}_{m_{\text{lin}}(\mathbf{x})}, \ \underbrace{\phi(\mathbf{x})^\top \mathbf{\Sigma}_{\text{lin}} \phi(\mathbf{x})}_{\kappa_{\text{lin}}(\mathbf{x},\mathbf{x})} + \Lambda^{-1}),$$
$$\text{where } \mathbf{\Sigma}_{\text{lin}}^{-1} := \sum_i \phi(\mathbf{x}_i) \, \Lambda \, \phi(\mathbf{x}_i)^\top + \delta \mathbf{I}_P. \tag{16}$$

We will make use of Eqs. 14 to 16 to write the mean and covariance function of the posterior approximation for neural networks, as shown in the next section.

## A.2  GP Posteriors from the Minimiser of Neural Networks

Khan et al. [16] derive GP predictive distributions for the minimisers of a variety of loss functions in Appendix B of their paper. We describe these below along with the resulting GP posteriors. Throughout, we denote a minimiser of the loss by $\mathbf{w}_*$.

**A regression loss:** For a regression loss function $\ell(y, f) := \frac{1}{2} \Lambda(y - f)^2$, they derive the following expression for the predictive distribution for the observations $y$ (see Equation 44, Appendix B.2 in their paper):

$$\hat{p}(y|\mathbf{x}, \mathcal{D}) := \mathcal{N}(y \,|\, f_{w_*}(\mathbf{x}), \ \mathbf{J}_{w_*}(\mathbf{x}) \mathbf{\Sigma}_* \mathbf{J}_{w_*}(\mathbf{x})^\top + \Lambda^{-1}),$$
$$\text{where } \mathbf{\Sigma}_*^{-1} := \sum_i \mathbf{J}_{w_*}(\mathbf{x}_i)^\top \Lambda \, \mathbf{J}_{w_*}(\mathbf{x}_i) + \delta \mathbf{I}_P. \tag{17}$$

We use $\hat{p}(y|\mathbf{x}, \mathcal{D})$ since this predictive distribution is not exact and is obtained using a type of Laplace approximation. Comparing this to Eq. 16, we can write the mean and covariance functions in a similar fashion as Eqs. 14 and 15:

$$m_{w_*}(\mathbf{x}) := f_{w_*}(\mathbf{x}), \quad \kappa_{w_*}(\mathbf{x}, \mathbf{x}') := \mathbf{J}_{w_*}(\mathbf{x}) \, \mathbf{\Sigma}_* \, \mathbf{J}_{w_*}(\mathbf{x}')^\top. \tag{18}$$

This is the result shown in Eq. 5 in Sec. 3.1.

**A binary classification loss:** A similar expression is available for binary classification with $y \in \{0, 1\}$, considering the loss $\ell(y, f) := -y \log \sigma(f) - (1 - y) \log(1 - \sigma(f)) = -yf + \log(1 + e^f)$ where $\sigma(f) := 1/(1 + e^{-f})$ is the sigmoid function. See Equation 48, Appendix B.2 in Khan et al. [16]. The predictive distribution is given as follows:

$$\hat{p}(y|\mathbf{x}, \mathcal{D}) := \mathcal{N}(y \,|\, \sigma(f_{w_*}(\mathbf{x})), \; \Lambda_{w_*}(\mathbf{x}) \, \mathbf{J}_{w_*}(\mathbf{x}) \, \mathbf{\Sigma}_* \, \mathbf{J}_{w_*}(\mathbf{x})^\top \Lambda_{w_*}(\mathbf{x}) + \Lambda_{w_*}(\mathbf{x})),$$

$$\text{where } \mathbf{\Sigma}_*^{-1} := \sum_i \mathbf{J}_{w_*}(\mathbf{x}_i)^\top \, \Lambda_{w_*}(\mathbf{x}_i) \, \mathbf{J}_{w_*}(\mathbf{x}_i) + \delta \mathbf{I}_P. \tag{19}$$

where $\mathbf{\Lambda}_{w_*}(\mathbf{x}) := \sigma(f_{w_*}(\mathbf{x})) [1 - \sigma(f_{w_*}(\mathbf{x}))]$. The predictive distribution does not respect the fact that $y$ is binary and treats it like a Gaussian. This makes it comparable to Eq. 16. Comparing the two, we can conclude that the above corresponds to the predictive posterior distribution of a GP regression model with $y = f(\mathbf{x}) + \epsilon$ where $\epsilon \sim \mathcal{N}(0, \Lambda_{w_*}(\mathbf{x}))$ with the mean and covariance function as shown below:

$$m_{w_*}(\mathbf{x}) := \sigma(f_{w_*}(\mathbf{x})), \quad \kappa_{w_*}(\mathbf{x}, \mathbf{x}') := \Lambda_{w_*}(\mathbf{x}) \, \mathbf{J}_{w_*}(\mathbf{x}) \, \mathbf{\Sigma}_* \, \mathbf{J}_{w_*}(\mathbf{x}')^\top \Lambda_{w_*}(\mathbf{x}). \tag{20}$$

This is the result used in Eq. 10 in Sec. 3.4 for binary classification. A difference here is that the mean function is passed through the sigmoid function and the covariance function has $\Lambda_{w_*}(\mathbf{x})$ multiplied on the both sides. These changes appear because of the nonlinearity in the loss function introduced due to the sigmoid link function.

**A multiclass classification loss:** The above result straightforwardly extends to the multiclass classification case by using multinomial-logit likelihood (or softmax function). For this the loss can be written as follows:

$$\ell(\mathbf{y}, \mathbf{f}) = -\mathbf{y}^\top \mathcal{S}(\mathbf{f}) + \log \left( 1 + \sum_{k=1}^{K-1} e^{f_k} \right), \quad \text{where } k\text{'th element of } \mathcal{S}(\mathbf{f}) \text{ is } \frac{e^{f_j}}{1 + \sum_{k=1}^{K-1} e^{f_k}} \tag{21}$$

where the number of categories is equal to $K$, $\mathbf{y}$ is a one-hot-encoding vector of size $K - 1$, $\mathbf{f}$ is $K - 1$ length output of the neural network, and $\mathcal{S}(\mathbf{f})$ is the softmax operation which maps a $K - 1$ length real vector to a $K - 1$ dimensional vector with entries in the open interval $(0, 1)$. The encoding in $K - 1$ length vectors ignores the last category which then ensures identifiability [35]. In a similar fashion to the binary case, the predictive distribution of the $K - 1$ length output $\mathbf{y}$ for an input $\mathbf{x}$ can be written as follows:

$$\hat{p}(\mathbf{y}|\mathbf{x}, \mathcal{D}) := \mathcal{N}(\mathbf{y} \,|\, \mathcal{S}(\mathbf{f}_{w_*}(\mathbf{x})), \; \mathbf{\Lambda}_{w_*}(\mathbf{x}) \, \mathbf{J}_{w_*}(\mathbf{x}) \, \mathbf{\Sigma}_* \, \mathbf{J}_{w_*}(\mathbf{x})^\top \mathbf{\Lambda}_{w_*}(\mathbf{x})^\top + \mathbf{\Lambda}_{w_*}(\mathbf{x})),$$

$$\text{where } \mathbf{\Sigma}_*^{-1} := \sum_i \mathbf{J}_{w_*}(\mathbf{x}_i)^\top \, \mathbf{\Lambda}_{w_*}(\mathbf{x}_i) \, \mathbf{J}_{w_*}(\mathbf{x}_i) + \delta \mathbf{I}_P. \tag{22}$$

where $\mathbf{\Lambda}_{w_*}(\mathbf{x}) := \mathcal{S}(\mathbf{f}_{w_*}(\mathbf{x})) [1 - \mathcal{S}(\mathbf{f}_{w_*}(\mathbf{x}))]^\top$ is a $(K - 1) \times (K - 1)$ matrix and $\mathbf{J}_{w_*}(\mathbf{x})$ is the $(K - 1) \times P$ Jacobian matrix. The mean function in this case is a $K - 1$ length matrix and the covariance function is a square matrix of size $K - 1$. Their expressions are shown below:

$$\mathbf{m}_{w_*}(\mathbf{x}) := \mathcal{S}(\mathbf{f}_{w_*}(\mathbf{x})), \quad \mathbf{K}_{w_*}(\mathbf{x}, \mathbf{x}') := \mathbf{\Lambda}_{w_*}(\mathbf{x}) \, \mathbf{J}_{w_*}(\mathbf{x}) \, \mathbf{\Sigma}_* \, \mathbf{J}_{w_*}(\mathbf{x}')^\top \mathbf{\Lambda}_{w_*}(\mathbf{x}'). \tag{23}$$

**General case:** The results above hold for a generic loss function derived from a generalised linear model (GLM) with an invertible function $\mathbf{h}(\mathbf{f})$, e.g., $\ell(\mathbf{y}, \mathbf{f}) := -\log p(\mathbf{y}|\mathbf{h}(\mathbf{f}))$. For example, for a Bernoulli distribution, the link function $h(f)$ is equal to $\sigma$. In the GLM literature, $\mathbf{h}^{-1}$ is known as the link function. Given such a loss, the only quantity that changes in the above calculations is $\mathbf{\Lambda}_{w_*}(\mathbf{x}, \mathbf{y}) := \nabla_{ff}^2 \ell(\mathbf{y}, \mathbf{f})$, which is the second derivative of the loss with respect to $\mathbf{f}$, and might depend both on $\mathbf{x}$ and $\mathbf{y}$.

## A.3  GP Posterior from the Iterations of a Neural-Network Optimiser

The results of the previous section hold only at a minimiser $\mathbf{w}_*$. Khan et al. [16] generalise this to iterations of optimisers. They did this for a variational inference algorithm and also for its deterministic version that resembles RMSprop. We now describe these two versions. We will only consider binary classification using the setup described in the previous section. The results can be easily generalised to multiclass classification.

**GP posterior from iterations of a variational inference algorithm:** Given a Gaussian variational approximation $q_j(\mathbf{w}) := \mathcal{N}(\mathbf{w}|\boldsymbol{\mu}_j, \boldsymbol{\Sigma}_j)$ at iteration $j$, Khan et al. [16] used a natural-gradient variational inference algorithm called the variational-online Newton (VON) method [15]. Given a $q_j(\mathbf{w})$, the algorithm proceeds by first sampling $\mathbf{w}_j \sim q_j(\mathbf{w})$, and then updating the variational distribution. Surprisingly, the procedure used to derive a GP predictive distribution for the minimiser generalises to this update too. An expression for the predictive distribution is given below:

$$\hat{p}_{j+1}(y|\mathbf{x}, \mathcal{D}) := \mathcal{N}(y \mid \sigma(f_{w_j}(\mathbf{x})), \ \Lambda_{w_j}(\mathbf{x}) \, \mathbf{J}_{w_j}(\mathbf{x}) \, \boldsymbol{\Sigma}_j \, \mathbf{J}_{w_j}(\mathbf{x})^\top \Lambda_{w_j}(\mathbf{x}) + \Lambda_{w_j}(\mathbf{x})^{-1}), \quad (24)$$

$$\text{where } \boldsymbol{\Sigma}_{j+1}^{-1} := (1 - \beta_j)\boldsymbol{\Sigma}_j^{-1} + \beta_j \left[ \sum_i \mathbf{J}_{w_j}(\mathbf{x}_i)^\top \Lambda_{w_j}(\mathbf{x}_i) \, \mathbf{J}_{w_j}(\mathbf{x}_i) + \delta \mathbf{I}_P \right], \quad (25)$$

$$\boldsymbol{\mu}_{j+1} := \boldsymbol{\mu}_j - \beta_j \boldsymbol{\Sigma}_{j+1} \left[ N \nabla_w \bar{\ell}(\mathbf{w}_j) + \delta \boldsymbol{\mu}_j \right], \quad (26)$$

where $\bar{\ell}(\mathbf{w}) := \frac{1}{N} \sum_{i=1}^N \ell(y_i, f_w(\mathbf{x}_i))$. The predictive distribution takes the same form as before, but now the covariance and mean are updated according to the VON updates. The VON updates are essential to ensure the validity of the GP posterior, however, as Khan et al. [16] discuss, the RMSprop/Adam have similar update which enable us to apply the above results even when running such algorithms. We describe this next.

**GP posterior from iterations of RMSprop/Adam:** Khan et al. [16] propose a deterministic version of the above update where $\mathbf{w}_j$ is not sampled from $q_j(\mathbf{w})$ rather is set to be equal to $\boldsymbol{\mu}_j$, i.e., $\mathbf{w}_j = \boldsymbol{\mu}_j$. This gives rise to the following update:

$$\boldsymbol{\Sigma}_{j+1}^{-1} \leftarrow (1 - \beta_j)\boldsymbol{\Sigma}_j^{-1} + \beta_j \left[ \sum_i \mathbf{J}_{w_j}(\mathbf{x}_i)^\top \Lambda_{w_j}(\mathbf{x}_i) \, \mathbf{J}_{w_j}(\mathbf{x}_i) + \delta \mathbf{I}_P \right], \quad (27)$$

$$\mathbf{w}_{j+1} \leftarrow \mathbf{w}_j - \beta_j \boldsymbol{\Sigma}_{j+1} \left[ N \nabla_w \bar{\ell}(\mathbf{w}_j) + \delta \mathbf{w}_j \right], \quad (28)$$

with the variational approximation defined as $q_j(\mathbf{w}) := \mathcal{N}(\mathbf{w}|\mathbf{w}_j, \boldsymbol{\Sigma}_j)$. The form of the predictive distribution remains the same as Eq. 24.

As discussed in Khan et al. [15], the above algorithm can be made similar to RMSprop by using a diagonal covariance. By reparameterising the diagonal of $\boldsymbol{\Sigma}^{-1}$ as $\mathbf{s} + \delta \mathbf{1}$ where $\mathbf{s}$ is an unknown vector, we can rewrite the updates to update $\boldsymbol{\mu}$ and $\mathbf{s}$. This can then be written in a form similar to RMSprop as shown below:

$$\mathbf{s}_{j+1} \leftarrow (1 - \beta_j)\mathbf{s}_j + \beta_j \left[ \sum_i \Lambda_{w_j}(\mathbf{x}_i) \left[ \mathbf{J}_{w_j}(\mathbf{x}_i) \circ \mathbf{J}_{w_j}(\mathbf{x}_i) \right]^\top \right] \quad (29)$$

$$\mathbf{w}_{j+1} \leftarrow \mathbf{w}_j - \beta_t \frac{1}{\mathbf{s}_{j+1} + \delta \mathbf{1}} \circ \left[ N \nabla_w \bar{\ell}(\mathbf{w}_t) + \delta \mathbf{w}_j \right], \quad (30)$$

where $\circ$ defines element-wise product of two vectors, and the diagonal of $\boldsymbol{\Sigma}_{j+1}^{-1}$ is equal to $(\mathbf{s}_{j+1} + \delta \mathbf{1})$. This algorithm differs from RMSprop in two ways. First, the scale vector $\mathbf{s}_j$ is updated using the sum of the square of the Jacobians instead of the square of the mini-batch gradients. Second, there is no square-root in the preconditioner for the gradient in the second line. This algorithm is the diagonal version of the Online Generalised Gauss-Newton (OGGN) algorithm discussed in Khan et al. [16].

In practice, we ignore these two differences and employ the RMSprop/Adam update instead. As a consequence the variance estimates might not be very good during the iteration, even though the fixed-point of the algorithm is not changed [15]. This is the price we pay for the convenience of using RMSprop/Adam. We correct the approximation after convergence of the algorithm by recomputing the diagonal of the covariance according to Eq. 29. Denoting the converged solution by $\mathbf{w}_*$, we compute the diagonal $\mathbf{v}_*$ of the covariance $\boldsymbol{\Sigma}_*$ as shown below:

$$\mathbf{v}_* = \mathbf{1} / \left[ \delta \mathbf{1} + \sum_{i=1}^N \Lambda_{w_*}(\mathbf{x}_i) \left[ \mathbf{J}_{w_*}(\mathbf{x}_i) \circ \mathbf{J}_{w_*}(\mathbf{x}_i) \right]^\top \right], \quad (31)$$

# B  Detailed Derivation of FROMP Algorithm

In this section, we provide further details on Sec. 3.3.

$$\mathcal{L}(q(\mathbf{w})) := \mathbb{E}_{q(w)}\left[\frac{N}{\tau}\bar{\ell}_t(\mathbf{w}) + \log q(\mathbf{w})\right] - \mathbb{E}_{\tilde{q}_{w_t}(\mathrm{f})}\left[\log \tilde{q}_{w_{t-1}}(\mathbf{f})\right],$$

$$\text{where } \mathbf{w}_t \sim q(\mathbf{w}) \text{ and } \mathbf{w}_{t-1} \sim q_{t-1}(\mathbf{w}). \qquad (32)$$

Optimising this objective requires us to obtain the GP posterior $\tilde{q}_{w_t}(\mathbf{f})$. This can be easily done applying the DNN2GP result from Eq. 24 to this loss function. The VON update for the objective above takes the following form:

$$\mathbf{\Sigma}^{-1} \leftarrow (1-\beta)\mathbf{\Sigma}^{-1} + \beta\left[\sum_i \mathbf{J}_{w_t}(\mathbf{x}_i)^\top \Lambda_{w_t}(\mathbf{x}_i)\,\mathbf{J}_{w_t}(\mathbf{x}_i) - 2\nabla_\Sigma \mathbb{E}_{\tilde{q}_{w_t}(\mathrm{f})}\left[\log \tilde{q}_{w_{t-1}}(\mathbf{f})\right]\right], \quad (33)$$

$$\boldsymbol{\mu} \leftarrow \boldsymbol{\mu} - \beta\mathbf{\Sigma}\left[\frac{N}{\tau}\nabla_w\bar{\ell}_t(\mathbf{w}_t) - \nabla_\mu \mathbb{E}_{\tilde{q}_{w_t}(\mathrm{f})}\left[\log \tilde{q}_{w_{t-1}}(\mathbf{f})\right]\right]. \qquad (34)$$

where $\bar{\ell}_t(\mathbf{w}) := \frac{1}{N}\sum_{i\in\mathcal{D}_t}\ell(y_i, f_w(\mathbf{x}_i))$ and we have ignored the iteration subscript to simplify notation.

Using the $\boldsymbol{\mu}$ and $\mathbf{\Sigma}$ obtained with this iteration, we can define the following GP predictive posterior at a sample $\mathbf{w}_t \sim q(\mathbf{w})$:

$$\hat{p}_t(y|\mathbf{x}, \mathcal{D}) := \mathcal{N}(y\,|\,\sigma(f_{w_t}(\mathbf{x})),\ \Lambda_{w_t}(\mathbf{x})\,\mathbf{J}_{w_t}(\mathbf{x})\,\mathbf{\Sigma}\,\mathbf{J}_{w_t}(\mathbf{x})^\top \Lambda_{w_t}(\mathbf{x}) + \Lambda_{w_t}(\mathbf{x})^{-1}), \qquad (35)$$

Comparing this to Eq. 24, we can write the mean and covariance function as follows:

$$m_{w_t}(\mathbf{x}) := \sigma(f_{w_t}(\mathbf{x})), \quad \kappa_{w_t}(\mathbf{x}, \mathbf{x}') := \Lambda_{w_t}(\mathbf{x})\,\mathbf{J}_{w_t}(\mathbf{x})\,\mathbf{\Sigma}\,\mathbf{J}_{w_t}(\mathbf{x}')^\top \Lambda_{w_t}(\mathbf{x}). \qquad (36)$$

The mean vector obtained by concatenating $m_{w_t}(\mathbf{x})$ at all $\mathbf{x} \in \mathcal{M}$ is denoted by $\mathbf{m}_t$. Similarly, the covariance matrix $\mathbf{K}_t$ is defined as the matrix with $ij$'th entry as $\kappa_{w_t}(\mathbf{x}_i, \mathbf{x}_j)$. The corresponding mean and covariance obtained from samples from $q_{t-1}(\mathbf{w})$ are denoted by $\mathbf{m}_{t-1}$ and $\mathbf{K}_{t-1}$.

Given these quantities, the functional regularisation term has an analytical expression given as follows:

$$\mathbb{E}_{\tilde{q}_{w_t}(\mathrm{f})}\left[\log \tilde{q}_{w_{t-1}}(\mathbf{f})\right] = -\frac{1}{2}\left[\mathrm{Tr}(\mathbf{K}_{t-1}^{-1}\mathbf{K}_t) + (\mathbf{m}_t - \mathbf{m}_{t-1})^\top \mathbf{K}_{t-1}^{-1}(\mathbf{m}_t - \mathbf{m}_{t-1})\right], \qquad (37)$$

correct to a constant. Our goal is to obtain the derivative of this term with respect to $\boldsymbol{\mu}$ and $\mathbf{\Sigma}$. Both $\mathbf{m}_t$ and $\mathbf{K}_t$ are functions of $\boldsymbol{\mu}$ and $\mathbf{\Sigma}$ through the sample $\mathbf{w}_t = \boldsymbol{\mu} + \mathbf{\Sigma}^{1/2}\boldsymbol{\epsilon}$ where $\boldsymbol{\epsilon} \sim \mathcal{N}(0, \mathbf{I})$. Therefore, we can compute these derivative using the chain rule.

We note that the resulting algorithm is costly for large problems, and propose five approximations to reduce the computation cost, as described below.

**Approximation 1:** Instead of sampling $\mathbf{w}_{t-1}$, we set $\mathbf{w}_{t-1} = \boldsymbol{\mu}_{t-1}$ which is the mean of the posterior approximation $q_{t-1}(\mathbf{w})$ until task $t-1$. Therefore, we replace $\mathbb{E}_{\tilde{q}_{w_t}(\mathrm{f})}\left[\log \tilde{q}_{w_{t-1}}(\mathbf{f})\right]$ by $\mathbb{E}_{\tilde{q}_{w_t}(\mathrm{f})}\left[\log \tilde{q}_{\mu_{t-1}}(\mathbf{f})\right]$. This affects the mean $\mathbf{m}_{t-1}$ and $\mathbf{K}_{t-1}$ in Eq. 37.

**Approximation 2:** When computing the derivation of the functional regulariser, we will ignore the derivative with respect to $\mathbf{K}_t$ and only consider $\mathbf{m}_t$. Therefore, the derivatives needed for the update in Eqs. 33 and 34 can be approximated as follows:

$$\nabla_\mu \mathbb{E}_{\tilde{q}_{w_t}(\mathrm{f})}\left[\log \tilde{q}_{w_{t-1}}(\mathbf{f})\right] \approx -\left[\nabla_\mu \mathbf{m}_t\right]\mathbf{K}_{t-1}^{-1}(\mathbf{m}_t - \mathbf{m}_{t-1}), \qquad (38)$$

$$\nabla_\Sigma \mathbb{E}_{\tilde{q}_{w_t}(\mathrm{f})}\left[\log \tilde{q}_{w_{t-1}}(\mathbf{f})\right] \approx -\left[\nabla_\Sigma \mathbf{m}_t\right]\mathbf{K}_{t-1}^{-1}(\mathbf{m}_t - \mathbf{m}_{t-1}). \qquad (39)$$

This avoids having to calculate complex derivatives (e.g., derivatives of Jacobians).

**Approximation 3:** Instead of using the full $\mathbf{K}_{t-1}$, we factorise it across tasks, i.e., we approximate it by a block-diagonal matrix containing the kernel matrix $\mathbf{K}_{t-1,s}$ for all past tasks $s$ as the diagonal. This makes the cost of inversion linear in the number of tasks.

**Approximation 4:** Similarly to Eqs. 27 and 28, we use a deterministic version of the VON update by setting $\mathbf{w}_t = \boldsymbol{\mu}$, which corresponds to setting the random noise $\boldsymbol{\epsilon}$ to zero in $\mathbf{w}_t = \boldsymbol{\mu} + \mathbf{\Sigma}^{1/2}\boldsymbol{\epsilon}$. This approximation simplifies the gradient computation in Eqs. 38 and 39, since now the gradient with respect to $\mathbf{\Sigma}$ is zero. For example, in the binary classification case, $m_\mu(\mathbf{x}) := \sigma(f_\mu(\mathbf{x}))$, which does

not depend on $\boldsymbol{\Sigma}$. The gradient of $\mathbf{m}_t$ with respect to $\boldsymbol{\mu}$ is given as follows using the chain rule (here $\mathbf{m}_{t,s}$ is the sub-vector of $\mathbf{m}_t$ corresponding to the task $s$).

$$\nabla_\mu \mathbf{m}_{t,s}[i] = \nabla_\mu \left[ \sigma \left( f_\mu(\mathbf{x}_i) \right) \right] = \Lambda_\mu(\mathbf{x}_i) \mathbf{J}_\mu(\mathbf{x}_i)^\top, \text{ where } \mathbf{x}_i \in \mathcal{M}_s, \tag{40}$$

and where the second equality holds for canonical link functions. With these simplifications, we can write the VON update as follows:

$$\boldsymbol{\Sigma}^{-1} \leftarrow (1-\beta)\boldsymbol{\Sigma}^{-1} + \beta \left[ \sum_i \mathbf{J}_\mu(\mathbf{x}_i)^\top \Lambda_\mu(\mathbf{x}_i) \mathbf{J}_\mu(\mathbf{x}_i) \right], \tag{41}$$

$$\boldsymbol{\mu} \leftarrow \boldsymbol{\mu} - \beta\boldsymbol{\Sigma} \left[ \frac{N}{\tau} \nabla_\mu \bar{\ell}_t(\boldsymbol{\mu}) + \sum_{s=1}^{t-1} [\nabla_\mu \mathbf{m}_{t,s}] \mathbf{K}_{t-1,s}^{-1}(\mathbf{m}_{t,s} - \mathbf{m}_{t-1,s}) \right]. \tag{42}$$

**Approximation 5:** Similarly to Eqs. 29 and 30, our final approximation is to use a diagonal covariance $\boldsymbol{\Sigma}$ and replace the above update by an RMSprop-like update where we denote $\boldsymbol{\mu}$ by $\mathbf{w}$:

$$\mathbf{s} \leftarrow (1-\beta)\mathbf{s} + \beta \left[ \sum_i \Lambda_w(\mathbf{x}_i) \left[ \mathbf{J}_w(\mathbf{x}_i) \circ \mathbf{J}_w(\mathbf{x}_i) \right]^\top \right], \tag{43}$$

$$\mathbf{w} \leftarrow \mathbf{w} - \beta \frac{1}{\mathbf{s} + \delta\mathbf{1}} \circ \left[ \frac{N}{\tau} \nabla_w \bar{\ell}_t(\mathbf{w}) + \sum_{s=1}^{t-1} [\nabla_w \mathbf{m}_{t,s}] \mathbf{K}_{t-1,s}^{-1}(\mathbf{m}_{t,s} - \mathbf{m}_{t-1,s}) \right], \tag{44}$$

where we have added a regulariser $\delta$ to $\mathbf{s}$ in the second line to avoid dividing by zero. Previously [15], this regulariser was the prior precision. Ideally, when using a functional prior, we would replace this by another term. However, this term was ignored by making Approximation 4, and we use $\delta$ instead. The final Gaussian approximation is obtained with the mean equal to $\mathbf{w}$ and covariance is equal to a diagonal matrix with $1/(\mathbf{s} + \delta\mathbf{1})$ as its diagonal.

It is easy to see that the solutions found by this algorithm is the fixed point of this objective:

$$\min_w N\bar{\ell}_t(\mathbf{w}) + \tfrac{1}{2}\tau \sum_{s=1}^{t-1} (\mathbf{m}_{t,s} - \mathbf{m}_{t-1,s})^\top \mathbf{K}_{t-1,s}^{-1}(\mathbf{m}_{t,s} - \mathbf{m}_{t-1,s}), \tag{45}$$

Ultimately, this is an approximation of the objective given in Eq. 32, and is computationally cheaper to optimise.

We follow the recommendations of Khan et al. [16] and use RMSprop/Adam instead of Eqs. 27 and 28. This algorithm still optimises the objective given in Eq. 45, but the estimate of the covariance is not accurate. We correct the approximation after convergence of the algorithm by recomputing the diagonal of the covariance according to Eq. 43. Denoting the converged solution by $\mathbf{w}_*$, we compute the diagonal $\mathbf{v}_*$ of the covariance $\boldsymbol{\Sigma}_*$ as shown below:

$$\mathbf{v}_* = \mathbf{1} / \left[ \delta\mathbf{1} + \sum_{i=1}^{N} \Lambda_{w_*}(\mathbf{x}_i) \left[ \mathbf{J}_{w_*}(\mathbf{x}_i) \circ \mathbf{J}_{w_*}(\mathbf{x}_i) \right]^\top \right], \tag{46}$$

## C  Multiclass setting

When there are more than two classes per task, we need to use multiclass versions of the equations presented so far. We still make the same approximations as described in App. B.

**Reducing Complexity in the Multiclass setting:** We could use the full multiclass version of the GP predictive (Eq. 22), but this is expensive. To keep computational complexity low, we employ an individual GP over each of the $K$ classes seen in a previous task, and treat the GPs as independent.

We have $K$ separate GPs. Let $\mathbf{y}^{(k)}$ be the $k$-th item of $\mathbf{y}$. Then the predictive distribution over each $\mathbf{y}^{(k)}$ for an input $\mathbf{x}$ is:

$$\hat{p}(\mathbf{y}^{(k)}|\mathbf{x}, \mathcal{D}) := \mathcal{N}\big(\mathbf{y}^{(k)} \,|\, \mathcal{S}(\mathbf{f}_{w_*}(\mathbf{x}))^{(k)}, \;\; \boldsymbol{\Lambda}_{w_*}(\mathbf{x})^{(k)} \mathbf{J}_{w_*}(\mathbf{x}) \boldsymbol{\Sigma}_* \mathbf{J}_{w_*}(\mathbf{x})^\top \boldsymbol{\Lambda}_{w_*}(\mathbf{x})^{(k)\top}$$
$$+ \Lambda_{w_*}(\mathbf{x})^{(k,k)}\big), \quad (47)$$

where $\mathcal{S}(\mathbf{f}_{w_*}(\mathbf{x}))^{(k)}$ is the k-th output of the softmax function, $\mathbf{\Lambda}_{w_*}(\mathbf{x})^{(k)}$ is the $k$-th row of the Hessian matrix and $\Lambda_{w_*}(\mathbf{x})^{(k,k)}$ is the $k, k$-th element of the Hessian matrix. The Jacobians $\mathbf{J}_{w_*}(\mathbf{x})$ are now of size $K \times P$. Note that we have allowed $\mathcal{S}$ and $\Lambda_{w_*}(\mathbf{x})$ to be of size $K$ instead of $K - 1$. This is because we are treating the $K$ GPs separately.

The kernel matrix $\mathbf{K}_{t-1}$ is now a block diagonal matrix for each previous task's classes. This allows us to only compute inverses of each block diagonal (size $M \times M$), repeated for each class in each past task ($K(t-1)$ times), where $M$ is the number of memorable past examples in each task. This changes computational complexity to be linear in the number of classes per task, $K$, compared to Sec. 3.4 (which has analysis for binary classification for each task).

When choosing a memorable past (the subset of points to regularise function values over) for the logistic regression case, we can simply sort the $\Lambda_{w_*}(\mathbf{x}_i)$'s for all $\{\mathbf{x}_i\} \in \mathcal{D}_t$ and pick the largest, as explained in Sec. 3.2. In the multiclass case, these are now $K \times K$ matrices $\mathbf{\Lambda}_{w_*}(\mathbf{x}_i)$. We instead sort by $\mathrm{Tr}(\mathbf{\Lambda}_{w_*}(\mathbf{x}_i))$ to select the memorable past examples.

**FROMP for multiclass classification:** The solutions found by the multiclass algorithm is the fixed point of this objective (compare with Eq. 45):

$$\min_w N\bar{\ell}_t(\mathbf{w}) + \tfrac{1}{2}\tau \sum_{s=1}^{t-1} \sum_{k \in C_s} (\mathbf{m}_{t,s,k} - \mathbf{m}_{t-1,s,k})^\top \mathbf{K}_{t-1,s,k}^{-1} (\mathbf{m}_{t,s,k} - \mathbf{m}_{t-1,s,k}), \qquad (48)$$

where we define $C_s$ as the set of classes $k$ seen in previous task $s$, $\mathbf{m}_{t,s,k}$ is the vector of $m_{w_t}(\mathbf{x})$ for class $k$ evaluated at the memorable points $\{\mathbf{x}_i\} \in \mathcal{M}_s$, $\mathbf{m}_{t-1,s,k}$ is the vector of $m_{w_{t-1}}(\mathbf{x})$ for class $k$, and $\mathbf{K}_{t-1,s,k}$ is the kernel matrix from the previous task just for class $k$, always evaluated over just the memorable points from previous task $s$. By decomposing the last term over individual outputs and over the memorable past from each task, we have reduced the computational complexity per update.

# D   Functional prior approximation

We discuss why replacing weight space integral by a function space integral, as done below, results in an approximation:

$$\mathbb{E}_{q(w)}[\log q_{t-1}(\mathbf{w})] \approx \mathbb{E}_{\tilde{q}_{w_t}(f)}\left[\log \tilde{q}_{w_{t-1}}(\mathbf{f})\right],$$

A change of variable in many cases results in an equality, e.g., for $\mathbf{f} = \mathbf{X}\mathbf{w}$ with a matrix $\mathbf{X}$ and given any function $h(\mathbf{f})$, we can express the weight space integral as the function space integral:

$$\int h(\mathbf{X}\mathbf{w})\mathcal{N}(\mathbf{w}|\boldsymbol{\mu}, \boldsymbol{\Sigma})d\mathbf{w} = \int h(\mathbf{f})\mathcal{N}(\mathbf{f}|\mathbf{X}\boldsymbol{\mu}, \mathbf{X}\boldsymbol{\Sigma}\mathbf{X}^\top)d\mathbf{f}. \qquad (49)$$

Unfortunately, $\log q_{t-1}(\mathbf{w})$ can not always be written as a function of $\mathbf{f} := \mathbf{J}_{w_t}\mathbf{w}$. Therefore, the change of variable does not result in an equality. For our purpose, as long as the approximations provide a reasonable surrogate for optimisation, the approximation is not expected to cause issues.

# E   Further details on continual learning metrics reported

We report a backward transfer metric and a forward transfer metric on Split CIFAR (higher is better for both). The backward transfer metric is exactly as defined in Lopez-Paz and Ranzato [20]. The forward transfer metric is a measure of how well the method uses previously seen knowledge to improve classification accuracy on newly seen tasks. Let there be a total of $T$ tasks. Let $R_{i,j}$ be the classification accuracy of the model on task $t_j$ after training on task $t_i$. Let $R_i^{\mathrm{ind}}$ be the classification accuracy of an independent model trained only on task $i$. Then,

$$\text{Backward Transfer, BWT} = \frac{1}{T-1}\sum_{i=1}^{T-1} R_{T,i} - R_{i,i},$$

$$\text{Forward Transfer, FWT} = \frac{1}{T-1}\sum_{i=2}^{T} R_{i,i} - R_i^{\mathrm{ind}}.$$

FROMP achieves $6.1 \pm 0.7\%$, a much higher value compared to $0.17 \pm 0.9\%$ obtained with EWC and $1.8 \pm 3.1\%$ with VCL+coresets. For backward transfer, we used the BWT metric defined in [20] which roughly captures the difference in accuracy obtained when a task is first trained and its accuracy after the final task. Again, higher is better and quantifies the gain obtained with the future tasks. Here, FROMP has a score of $-2.6 \pm 0.9\%$, which is comparable to EWC's score of $-2.3 \pm 1.4\%$ but better than VCL+coresets which obtains $-9.2 \pm 1.8\%$.

Table 2: Summary of metrics on Split CIFAR. FROMP outperforms the baselines EWC and VCL+coresets. All methods are run five times, with mean and standard deviation reported.

| Method | Final average accuracy | Forward transfer | Backward transfer |
|---|---|---|---|
| EWC | $71.6 \pm 0.9\%$ | $0.17 \pm 0.9\%$ | $-2.3 \pm 1.4\%$ |
| VCL+coresets | $67.4 \pm 1.4\%$ | $1.8 \pm 3.1\%$ | $-9.2 \pm 1.8\%$ |
| FROMP | $\mathbf{76.2} \pm 0.4\%$ | $\mathbf{6.1} \pm 0.7\%$ | $-\mathbf{2.6} \pm 0.9\%$ |

# F   Further details on experiments

## F.1   Permuted MNIST

We use the Adam optimiser [17] with Adam learning rate set to 0.001 and parameter $\beta_1 = 0.99$, and also employ gradient clipping. The minibatch size is 128, and we learn each task for 10 epochs. We use $\tau = 0.5N$ for all algorithms, with 200 memorable points: FROMP, FRORP, FROMP-$L_2$ and FRORP-$L_2$. We use a fully connected single-head network with two hidden layers, each consisting of 100 hidden units with ReLU activation functions. We report performance after 10 tasks.

## F.2   Split MNIST

We use the Adam optimiser [17] with Adam learning rate set to 0.0001 and parameter $\beta_1 = 0.99$, and also employ gradient clipping. The minibatch size is 128, and we learn each task for 15 epochs. We use $\tau = 10N$ for all algorithms, with 40 memorable points: FROMP, FRORP, FROMP-$L_2$ and FRORP-$L_2$. We use a fully connected multi-head network with two hidden layers, each with 256 hidden units and ReLU activation functions.

**Smaller network architecture from Swaroop et al. [33].** Swaroop et al. [33] use a smaller network than the network we use for the results in Fig. 2a. They train VCL on a single-hidden layer network with 100 hidden units (and ReLU activation functions). To ensure faithful comparison, we reran FROMP (with 40 memorable points per task) on this smaller network, obtaining a mean and standard deviation over 5 runs of $(99.2 \pm 0.1)\%$. This is an improvement from Fig. 2a, which uses a larger network. We believe this is due to the pruning effect described in Swaroop et al. [33].

**Sensitivity to the value of $\tau$.** We tested FROMP and FROMP-$L_2$ with different values of the hyperparameter $\tau$. We found that $\tau$ can change by an order of magnitude without significantly affecting final average accuracy. Larger changes in $\tau$ led to greater than 0.1% loss in accuracy.

## F.3   Split CIFAR

We use the Adam optimiser [17] with Adam learning rate set to 0.001 and parameter $\beta_1 = 0.99$, and also employ gradient clipping. The minibatch size is 256, and we learn each task for 80 epochs. We use $\tau = 10N$ for all algorithms, with 200 memorable points: FROMP, FRORP, FROMP-$L_2$ and FRORP-$L_2$.

**Numerical results on Split CIFAR**. We run all methods 5 times and report the mean and standard error. For baselines, we train from scratch on each task and jointly on all tasks achieving $(73.6 \pm 0.4)\%$ and $(78.1 \pm 0.3)\%$, respectively. The final average validation accuracy of FROMP is $(76.2 \pm 0.4)\%$, FROMP-$L_2$ is $(75.6 \pm 0.4)\%$, SI is $(73.5 \pm 0.5)\%$ (result from Zenke et al. [37]), EWC is $(71.6 \pm 0.9)\%$, VCL + random coreset is $(67.4 \pm 1.4)\%$.

**Longer task sequence: 11 tasks of Split CIFAR.** We also run Split CIFAR for 11 tasks instead of the standard 6 tasks, and compare FROMP with FROMP-$L_2$ and FRORP for different sizes of

Figure 4: Results on Split CIFAR with 11 tasks as the number of memorable examples changes. A careful selection of memorable examples in FROMP gives better (/more consistent) results than random examples in FRORP, especially when the memory size is small.

Figure 5: This figure demonstrates our approach on a toy dataset. Figure (i) shows the result of training on the first task where memorable past examples are shown with big markers. These points usually are the ones that support the decision boundary. Figure (ii) shows the result after training on the second task where we see that the new network outputs are forced to give the same prediction on memorable past examples as the previous network. The new decision boundary classifies both task 1 and 2 well. Figure (iii) shows the result after training on five tasks, along with the memorable-past of each task. With our method, the performance over past tasks is maintained.

memorable past (Fig. 4). We find similar results to Fig. 3b in the main text, with FROMP typically out-performing FRORP, especially at smaller memorable sizes, but being similar to FROMP-$L_2$.

### F.4    Fewer memorable past examples

When we have fewer memorable past examples (for Figs. 3b and 3c), we increase $\tau$ to compensate for the fewer datapoints. For example, for Split CIFAR, when we have 40 memorable past examples per task (instead of 200), we use $\tau = (200/40) * 10N = 50N$ (instead of $\tau = 10N$ for 200 memorable past points).

## G    Toy data experiments

In this section, we use a 2D binary classification toy dataset with a small multi-layer perceptron to (i) demonstrate the brittleness and inconsistent behaviour of weight-regularisation, (ii) test FROMP's performance on different toy datasets of varying difficulty. As shown in Fig. 5 in App. G, we find that weight-regularisation methods like VCL (+coresets) perform much worse than functional-regularisation, with lower accuracy, higher variance over random seeds, and visually bad decision boundaries.

The toy dataset we use is shown in Fig. 5, along with how FROMP does well. In App. G.1, we show weight-space regularisation's inconsistent behaviour on this dataset, with results and visualisations. In App. G.2, we show that FROMP performs consistently across many variations of the dataset. Finally, hyperparameters for our experiments are presented in App. G.3. For all these experiments, we use a 2-hidden layer single-head MLP with 20 hidden units in each layer.

## G.1 Weight-space regularisation's inconsistent behaviour

Table 3: Train accuracy of FROMP, VCL (no coresets), VCL+coresets and batch-trained Adam (an upper bound on performance) on a toy 2D binary classification dataset, with mean and standard deviations over 5 runs for VCL and batch Adam, and 10 runs for FROMP. 'VCL' is without coresets. VCL-RP and FRORP have the same (random) coreset selections. VCL-MP is provided with 'ideal' coreset points as chosen by an independent run of FROMP. VCL (no coreset) does very poorly, forgetting previous tasks. VCL+coresets is brittle with high standard deviations, while FROMP is stable.

| FROMP | FRORP | VCL-RP | VCL-MP | VCL | Batch Adam |
|---|---|---|---|---|---|
| $99.6 \pm 0.2\%$ | $98.5 \pm 0.6\%$ | $92 \pm 10\%$ | $85 \pm 14\%$ | $68 \pm 8\%$ | $99.70 \pm 0.03\%$ |

Table 3 summarises the performance (measured by train accuracy) of FROMP and VCL+coresets on a toy dataset similar to that in Fig. 5. FROMP is very consistent, while VCL (with coresets) is extremely brittle: it can perform well sometimes (1 run out of 5), but usually does not (4 runs out of 5). This is regardless of the coreset points chosen for VCL. Note that coresets are chosen independently of training in VCL. Without coresets, VCL forgets many past tasks, with very low performance.

For VCL-MP, the coreset is chosen as the memorable past from an independent run of FROMP, with datapoints all on the task boundary. This selection of coreset is intuitively better than a random coreset selection. The results we show here are not specific to coreset selection. Any coreset selection (whether random or otherwise) all show the same inconsistency when VCL is trained with them. We provide visualisations of the brittleness of VCL in Fig. 6.

Figure 6: Three runs of VCL-MP on toy 2D data. These are the middle performing 3 runs out of 5 runs with different random seeds. VCL's inconsistent behaviour is clear.

## G.2 Dataset variations

Figs. 7 to 11 visualise the different dataset variations presented in Table 4. We pick the middle performing FROMP run (out of 5) and batch Adam run to show.

Table 4: Train accuracy of FROMP and batch-trained Adam (upper bound on performance) on variations of a toy 2D binary classification dataset, with mean and standard deviations over 10 runs (3 runs for Adam). FROMP performs well across variations. VCL (with coresets) performs significantly worse even on the original dataset ($92 \pm 10\%$). See App. G.2 for further experiments and for visualisations.

| Dataset variation | FROMP | Batch Adam |
|---|---|---|
| Original dataset | $99.6 \pm 0.2\%$ | $99.7 \pm 0.0\%$ |
| 10x less data (400 per task) | $99.9 \pm 0.0\%$ | $99.7 \pm 0.2\%$ |
| 10x more data (40000 per task) | $96.9 \pm 3.0\%$ | $99.7 \pm 0.0\%$ |
| Introduced 6th task | $97.8 \pm 3.3\%$ | $99.6 \pm 0.1\%$ |
| Increased std dev of each class distribution | $96.0 \pm 2.4\%$ | $96.9 \pm 0.4\%$ |
| 2 tasks have overlapping data | $90.1 \pm 0.8\%$ | $91.1 \pm 0.3\%$ |

Figure 7: FROMP (middle performing of 5 runs) and batch Adam on a dataset 10x smaller (400 points per task).

Figure 8: FROMP (middle performing of 5 runs), left, and batch Adam, right, on a dataset 10x larger (40,000 points per task).

### G.3 VCL and FROMP hyperparameter settings for toy datasets

**FROMP.** We optimised the number of epochs, Adam learning rate, and batch size. We optimised by running different hyperparameter settings for 5 runs on the toy dataset in Fig. 5, and picking the settings with largest mean train accuracy. We found the best settings were: number of epochs=50, batch size=20, learning rate=0.01. The hyperparameters were then fixed across all toy data experimental runs, including across dataset variations (number of epochs was appropriately scaled by 10 if dataset size was scaled by 10).

**VCL+coresets.** We optimised the number of epochs, the number of coreset epochs (because VCL+coresets trains on non-coreset data first, then on coreset data just before test-time: see Nguyen et al. [22]), learning rate (we use Adam to optimise the means and standard deviations of each parameter), batch size, and prior variance. We optimised by running various settings for 5 runs and picking the settings with largest mean train accuracy. We found the best settings were: number of epochs=200, number of coreset epochs=200, a standard normal prior (variance=1), batch size=40, learning rate=0.01. VCL is slow to run (an order of magnitude longer) compared to the other methods (FROMP and batch Adam).

### G.4 Importance of kernel being over all layer weights

In this section, we show the importance of using all weights of the neural network, instead of just the last layer. Our kernel is over all weights from all layers. We run the same toy experiment, and consider the entropies of the Gaussian distributions for weights in each layer. We plot the histogram of these entropies in Fig. 12. As can be seen, all layers have weights with high uncertainty (high entropy), especially for the first few tasks. Note that as we train for more tasks, we expect the uncertainties to reduce as our network parameters become more certain having seen more data.

Figure 9: FROMP (middle performing of 5 runs), left, and batch Adam, right, on a dataset with a new, easy, 6th task.

Figure 10: FROMP (middle performing of 5 runs), left, and batch Adam, right, on a dataset with increased standard deviations of each class' points, making classification tougher.

Therefore, by considering uncertainties across weights in all layers, instead of just the last layer, we might expect better performance.

## H   Task boundary detection

In this section, we consider the case where data is separated into tasks, but we are not provided task boundaries during training. Our goal is to detect the task boundaries. Many of the ideas in this section are inspired from Titsias et al. [34] Section 3.

We consider 10 tasks of Permuted MNIST, with minibatches arriving without task ID information. We wish to automatically detect when a new minibatch belongs to a new task. We use the same network and hyperparameters as in App. F.

The key insight is that, when we first see data from a new task, we expect this data to be far (in input space) from data we have observed so far. Therefore, predictions over this new data with our current network parameters, $\mathbf{m}_t$, should be similar to predictions with our prior network parameters, $\mathbf{m}_{t-1}$. This is in contrast to when we see data from the current task, when predictions with our current network parameters will be very different to our prior network parameters.

Using this insight, we perform a test on every new minibatch of data, in order to determine whether it is from a new task or not. This test is performed before training on the minibatch.

For every new minibatch, we:

1. Calculate $(m_{t,i} - m_{t-1,i})^2$ for each sample $i$ in the minibatch where $m_{t,i}$ and $m_{t-1,i}$ are predictive mean obtained using current and past networks respectively. When we see a new task, we expect this value to be small.

Figure 11: FROMP (middle performing of 5 runs), left, and batch Adam, right, on a dataset with 2 tasks having overlapping data, which is not separable.

Figure 12: Histogram of entropy of distribution the distribution of weights for different layers (row) and task (columns). For each layer, we take all weights and plot the histogram of their entropies. Left-most is after the first task, and right-most is after the last task. We see that the entopy is high across layers, implying that there is significant uncertainties about the weights for all of them, not only the last layer (layer 3 in this case).

2. Calculate Welch's t-test statistic between the current and the previous minibatch's samples. For the multi-class setup of Permuted MNIST, we repeat this for each function, and average this statistic across the functions.

3. If the statistic is sufficiently high (above a threshold), we detect a new task.

We find that this method is very good at determining task boundaries. We always successfully recognise a task change, with no mistakes, over a wide range of thresholds. Note that we do not conduct the test for the first 10 iterations of training on a new task.

We plot the Welch's t-test statistic between minibatches in Fig. 13 for a specific run. As can be seen, we can use a range of threshold values (approximately 0.9 to 1.8, limited by detecting the very first task change) to successfully recognise that the task has changed.

We found that using just mean predictions to be good enough for determining task boundaries in this setting. Ideally, in more complicated scenarios, we might want to use the full GP predictive distribution, and compare that to the predictive distribution from the GP prior. We could then use a divergence to determine how similar the two distributions are, with the expectation that new tasks have small divergence.

(a) Square of difference in mean predictions

(b) Welch's t-test statistic

Figure 13: Detecting task boundary changes in 10 tasks of Permuted MNIST. (a) The square of the difference in means reduces noticeably whenever a minibatch from a new task is seen for the first time. (b) We can perform Welch's t-test to detect these changes, and threshold on this value to detect a new task.

# I   Changes in the camera-ready version compared to the submitted version

- We expanded the Related Works section.
- We added the task boundary detection experiment (App. H).
- We ran Split CIFAR on 11 tasks (App. F).
- Added App. G.4 discussing (with a toy visualisation) the importance of using all layer weights in the kernel matrix, not just the last layer weights.
- We updated Fig. 3b and Fig. 3c in the main text using the newest hyperparameters we found. The new plot shows that FROMP and FROMP-$L_2$ are similar in Fig. 3b, but slightly further apart in Fig. 3c. The old figures are in Fig. 14, which we believe should be attainable with different hyperparameters.

# J   Author Contributions Statement

List of Authors: Pingbo Pan, Siddharth Swaroop, Alexander Immer, Runa Eschenhagen, Richard E. Turner, Mohammad Emtiyaz Khan.

P.P, S.S., and M.E.K. conceived the original idea of using DNN2GP for continual learning. This was then discussed with R.E., R.T., and A.I. The DNN2GP result from Section 3.1 is due to A.I. The memorable past method in Section 3.2 is due to M.E.K. The FROMP algorithm in Algorithm 1 was originally conceived by P.P., S.S. and M.E.K. The idea of functional prior was conceived by S.S. and M.E.K. Based on this idea, S.S. and M.E.K. wrote a derivation using the variational approach, which is currently written in Section 3.3. R.E., A.I. and R.T. regularly provided feedback for the main methods.

P.P. conducted all experiments, with feedback from M.E.K., A.I., R.E, and S.S. S.S. made corrections to some of the code, fixed hyperparameter reporting, and also did baseline comparisons.

(a) Split CIFAR      (b) Permuted MNIST

Figure 14: Previous figures for average accuracy with respect to the number of memorable examples.

The first version of the paper was written by M.E.K. with some help from the other authors. S.S revised the paper many times and also rewrote many new parts. Detailed derivation in Appendix is written by S.S. and M.E.K. The authors A.I., R.E. and R.T. provided feedback during the writing of the paper.

M.E.K. and S.S. led the project.