[Reviews · NeurIPS 2020]

Review 1

Summary and Contributions: This paper proposes a functional regularization approach (FROMP)for continual learning with the help of past memory. It extends the existing functional regularization method by converting the whole neural network into a Gaussian formulation.

Strengths: This work extend the previous method (FRCL) by regarding the whole network as gaussian formulation. At the same time, FROMP infers the gaussian process with a cheaper important sampling w.r.t. the uncertainty of each sample. It is possible the reason why FROMP require less samples to get better results. The mathematical formulation of the basic model is quite elegant. The algorithm require not too much extra cost for the functional regularization term. The experiment results are astonishing.

Weaknesses: The idea is interesting, but not novel enough. line 43"especially in the early stage..." can you explain the intuition or give some extra experiment to show this? What is the fundamental difference between converting the whole network and just converting the last layer, and how this effect the early stage training? could you please give some hit for the new form of the regularization term ($m_t - m_{t-1}$)(compared with FRCL) . What kind of role does it play in the learning process. The setting is still task bounded, is it possible to do tasks detection as mentioned in FRCL? Does the sampling according to $\Lambda$ really make sense? In Figure 2.a FROMP does not outperform FRORP too much. Figure 3.a denotes that FROMP only beat the SI when the accuracy is about 0.75. While in Figure 3.b we can see that its corresponding sample number is 70. At this moment, the random sampling does a better job. Can we infer that random sampling is good enough better performance? About time complexity. The complexity of computing $Lambda$ and sorting is not counted. ----------------------------------------------------------------------------------------- after rebuttal The author's feedback solves my confusion of this paper, and give a quite convincing answer to my concern on the experimental result. Since this paper is elegant solution of continual learning under the umbrella of functional regularization and the external storage, I would like to bump my score to 7 . This is paper should be published from my personal perspective.

Correctness: yes

Clarity: yes

Relation to Prior Work: yes

Reproducibility: Yes

Additional Feedback:


Review 2

Summary and Contributions: This paper proposes to develop an efficient functional regularization approach for supporting continual learning with deep models. The inherent problem of catastrophic forgetting is well-known and existing approach either use weight regularization strategies or simplified functional regularization methods. Building upon the DNN2GP framework, this work provides a new perspective to this problem and advocate the use of noise precision as a relevance measure for determining memorable samples from the past to perform the regularization.

Strengths: A very well written paper. Functional regularization is a more direct and effective strategy compared to weight regularization and the DNN2GP-based formulation is highly intuitive. By presenting clear comparisons to FRCL (both conceptually and empirically), the authors present a convincing argument. Though the approach builds upon the existing theoretical framework of DNN2GP, I think the work is sufficiently novel. The use of noise precision to select memorable samples is simple and computationally effective. Can the authors clarify (maybe even include an empirical result) how reliable this approach is under distribution shifts? Also in regard to ranking influential samples, how does this approach compare to simple gradient based sample ranking schemes (e.g. Deep Batch Active Learning by Diverse, Uncertain Gradient Lower Bounds, ICLR 2020).

Weaknesses: While the proposed idea and formulation are promising, the empirical results are limited. In particular, more empirical results (additional datasets, more complex incoming tasks) can be included to clearly demonstrate the improvements over random selection (FRORP) and using L2 (identity sigma). Also, including experiments with 10s of tasks (currently 6 is considered) will further clarify how effective the sample selection method is.

Correctness: To the best of my understanding, this is technically accurate and the empirical methodology is correct.

Clarity: Yes. It is clearly written and a good read.

Relation to Prior Work: Sufficient discussion on positioning this work in the context of the prior art.

Reproducibility: Yes

Additional Feedback: Update after author response: I am satisfied by the author's responses to my questions. The only suggestion I have is including a discussion on the connection between the proposed memorable sample selection procedure and existing gradient-based sample selection methods in the active learning literature.


Review 3

Summary and Contributions: The authors propose a FROMP, a novel method for Continual Learning based on a GP formulation of deep networks that elegantly combines rehearsal(memory)-based Continual Learning methods with Functional Regularisation, a recently proposed regularisation strategy possibly more promising than popular weight-space based regularisation methods. Experimental results are obtained on standard benchmarks with strong resutls.

Strengths: - The fact that FROML does not require solving a discrete optimisation problem is a big strength, as Titsias et. al suggest that this is a non-trivial problem with significant computational cost and various competing objective funcstions. - Memorable examples appear highly intuitive as shown Figure 1b), an improvement on Titsias et. al, for which it is less clear why certain examples are preferable over others as inducing points. - Results on considered experiments are consistently strong, adequate ablation studies are considered. - I was happy to see an explicit evaluation of Forward and Backward Transfer, a key metric unfortunately often not reported in CL publications. I suggest computing those results for all benchmarks and reporting them in the main text to hopefully help make this a standard comparison. Personally, I would also argue that the area under the learning curve might be a preferable measure for forward transfer, as it also counts the speed of learning as opposed to merely the final performance.

Weaknesses: - The submission would benefit from clarifying assumptions as early as possible to help categorise this work in the array of possible solutions to a practical CL problem. Specifically, as presented this is a competitive solution provided: 1. The use of memory is possible in an application of interest 2. Clear task boundaries exist and can be identified or are provided. - While the connections and differences to the most immediately related work (Section 1.1) are clearly described, I would have liked to see a boarder review of recent work in Continual Learning. Currently, references to some important past publications are scattered throughout the text (e.g. EWC and VCL in Section 2), which makes me wonder why a related work section was introduced. I suggest either expanding the related work section and moving the majority of discussion of previous work there, as well as including work not currently referenced. If the authors find the space limit in the main text constraining, I suggest moving a broader discussion in the Appendix. - While the presented results on Image datasets are good, the CL community has to start considering more challenging and realistic tasks to make impact on other areas of Machine Learning. There has been very little progress in the last 2-3 years in terms of convincing standard applications and benchmarks of Continual Learning, with current experimental protocols being hardly changed and primarily focused on Image classification. I would be happy to consider raising my score if the authors introduced an additional experiment on a challenging and convincing CL problem such as sequential decision making (e.g. Contextual Bandits or Reinforcement Learning).

Correctness: - Experiments seem to have been carried out to a high standard, closely following the protocol established in previous work with attention to important details.

Clarity: With exception of the related work, I would place this work in the top 15% of NeurIPS submissions in terms of clarity. Importantly, the authors provide sufficient background for readers not intimately familiar with the GP literature, which may be useful for the CL community. In addition to a clearly written main text, a comprehensive and complementary appendix is provided.

Relation to Prior Work: See my point in weaknesses. While I think the work is clearly delineated from closely related publications, readers new to CL would benefit from a broader overview of the field. This could be added with minimal work in the rebuttal.

Reproducibility: Yes

Additional Feedback: - What is the behaviour of the proposed algorithm as the number of memorable past examples is increased further? The slope in Figure 3 b) suggests that further significant improvements could be achieved with a larger memory. At which point does this saturate? Could an adequate number of examples be chosen prior to running a possibly expensive experiment? - How does FROMP perform in a CL setting with a large number of tasks? I suggest evaluating the method of the Sequential Omniglot benchmark proposed by Schwarz et al. (for which results for FRCL are available). As this is also an image-classification based CL problem, those results should be relatively easy to obtain using the existing codebase. - An interesting use of predictive uncertainty has been proposed in the closely related work by Titsias et al., using the predictive uncertainty for changepoint detection. I would be interested in seeing a similar experiment by the authors, which might provided some further insight into whether this is a promising approach. Instead of merely using the T-test proposed by the authors I suggest combining the resulting statistics with a classic changepoint detection algorithm such as (Adams, Ryan Prescott, and David JC MacKay. "Bayesian online changepoint detection." arXiv preprint arXiv:0710.3742 (2007).) --- Update: I'm very happy with the author's response and understand that some of my feedback about experiments of higher difficulty could not be incorporated in the short time authors had to write a rebuttal. However, I trust that the authors will include additional results they promised for a camera-ready version. I will therefore stick with my rating


Review 4

Summary and Contributions: A regularization based lifelong learning approach is proposed in the paper. The approach selects a set of memorable past examples that are near the decision boundary. When new tasks come, these memorable past examples are used to regularize the outputs to be similar instead of regularizing weights to be close.

Strengths: Instead of regularizing network weights, the proposed approach regularize network outputs instead. And memorable past examples are selected to be used to compute the regularization constraints. Experiments are conducted on Permuted and Split MNIST, and Split CIFAR.

Weaknesses: 1. The contribution is limited. What are the real contributions of the paper? The idea of regularizing the outputs (or functional-regularization) has already been explored, as already said in the paper. Combining the idea of regularizing the outputs with memory-based methods is also already explored. Please see GEM [1] and A-GEM [2]. What makes this approach better or important, e.g. how is the proposed approach better than GEM or A-GEM? [1]Gradient Episodic Memory for Continual Learning. [2] Efficient Lifelong Learning with A-GEM. 2. The related work section is very limited. It is encouraged to discuss more lifelong learning approached besides regularization-based ones, as well as the relations between the proposed approach and existing approaches more thoroughly. To list a few more papers: [1] Gradient Episodic Memory for Continual Learning. [2] Efficient Lifelong Learning with A-GEM. [3] Learning without Forgetting. [4] Memory aware synapses: Learning what (not) to forget. [5] Riemannian walk for incremental learning: Understanding forgetting and intransigence. [6] Progressive Neural Networks. [7] PathNet: Evolution channels gradient descent in super neural networks. 3. The number of tasks used in the experiment section is limited. With increasing number of tasks, is optimization constraints violated frequently similarly as in GEM? 4. Only very limited SOTA methods are compared against. It is hard to see the effectiveness of the proposed approach. As in GEM, when number of tasks is 5, there are several SOTA approaches with accuracy around 95%. 5. There is not a gap between FRORP and FROMP. Maybe random selection is enough, maybe a gap would appear with much more tasks. It is hard to say whether example selection as in Sec. 3.2 is necessary or not. -------------------------------------------------------------------------------------------- After reading the rebuttal and talking with other reviewers, I agree that when #samples to store is small, the gap between random selection and proposed approach is obvious. My concern towards comparison with GEM and A-GEM type of approach is addressed through the discussion. The author also promised to provide longer sequence of 11 tasks, and enrich the related work section. Therefore, I raised my rating to accept.

Correctness: Yes.

Clarity: The paper is well written and easy to follow.

Relation to Prior Work: It is unclear how the proposed approach contains key differences from previous works.

Reproducibility: Yes

Additional Feedback: No further comments, please see the weakness section for comments and suggestions.

[Author Response · NeurIPS 2020]

We thank the reviewers for their time and feedback.

# R1: Score 6 (confidence 3)

**Q1: "Not novel enough".** We disagree. The content of Sec. 3.2 and 3.3 are entirely new. Both R2 and R3 agree.

**Q2: "What is the fundamental difference between converting whole network vs only the last layer"?** In the beginning of training, weights are changing frequently from task to task, therefore their uncertainty is high. Using only the last layer ignores the uncertainty in the rest of the weights. This could hurt performance a lot in the beginning. We will add more explanation in the paper, and include a small illustrative experiment in the Appendix.

**Q3: "What role does the ... regularization term play ... compared with FRCL"?** This term ensures that the current DNN outputs (mean) after task $t$ remain close to the DNN after task $t-1$, which is desirable to avoid forgetting. In contrast, FRCL only optimise the 'variance information' to be close, which may not be as effective (see line 44).

**Q4: "Is it possible to do task detection?"** Yes, it is possible to do this in the same way as FRCL paper.

**Q5: "Does sampling according to $\Lambda$ really makes sense?"** Yes, this is closely related to Kernel methods, such as kernel ridge-regression and SVM, where this strategy enables us to pick boundary/high-leverage points. This also leads to an intuitive and computationally cheap method (both R2 and R3 agree).

**Q6: "FROMP does not outperform FRORP"... Is random sampling enough?** It is not enough, and a careful selection is required when memory size is small. See Split CIFAR results in Figures 3b and 3c. On MNIST, FRORP is similar to FROMP because the task is very simple, and very little is gained from a careful selection of memorable examples. We will add results for 11 tasks on Split CIFAR where we expect to see even more of a difference.

**Q7: The complexity of computing $\Lambda$.** It costs $O(N)$. We will clarify this in the paper.

---

# R2: Score 7 (confidence 3)

**Q8: Reliability under distribution shift:** This should not be an issue as long as the memory is large enough.

**Q9: "How does this compare to simple gradient based sample ranking schemes (e.g. Deep Batch Active learning)".** Very good point. Out method turns out to be similar to 'Conf' selection, but we arrived at it with a different approach. We will add this reference and add a discussion in the paper.

**Q10: Include experiment with 10s of tasks.** Yes, we will add this. Also see our answer to Q14.

---

# R3: Score 8 (confidence 4)

**Q11: Add Forward and Backward transfer to all results.** Yes, we will add it.

**Q12: "Add clarifying assumptions earlier to help categorise this work".** Yes, we will add it (around line 73).

**Q13: Broader review of recent work in Continual Learning.** We agree and will improve it. Due to page limit, our Related Work section is short. For camera-ready, we can devote around half a page to include many other related works.

**Q14: Ready to raise score if a challenging and convincing CL result is added.** Thanks for the suggestion. Since our main contribution is to propose a new method, we focused on standard benchmarks. We will add the Split CIFAR experiment with 11 tasks (previous papers used only 6 tasks), and the "predictive uncertainty for change-point detection" in the Appendix. Adding many more challenging experiments takes more time and space, and we will explore scalability (e.g., large classes and datasets) and generality (e.g. in RL and Bandits) of the method in a separate future work.

**Q15: Behaviour as number of memorable points increases.** Good question. The performance saturates as expected. It is not clear how to find the number of examples beforehand, but one could easily design an online, greedy scheme to determine this number.

---

# R4: Score 5 (confidence 4)

**Q16: "Limited contributions".** We disagree. The content of Sec. 3.2 and 3.3 are entirely new. Both R2 and R3 agree.

**Q17: Discuss more related approaches other than regularization ones.** Thanks. We will fix this (including adding all the references mentioned, and a comparison with GEM). Also, see Q13.

**Q18: "Is optimization constraints violated?"** This is a misunderstanding. There are no constraints in our approach.

**Q19: Gap between FRORP and FROMP.** This is a misunderstanding. Please see our answer to Q6.

[Meta-Review · NeurIPS 2020]

All four expert reviewers were positive about this work, and the author rebuttal along with a lively post-rebuttal discussion improved the opinions. I agree with the reviewers that this is a high quality paper and my decision is to accept. I encourage the authors to take reviewer suggestions into account -- especially the promise to provide longer task sequences and the discussion of connections between gradient-based sample selection and the proposed memorable sample selection approach.